# ReCode: Unify Plan and Action for Universal Granularity Control

## Abstract

Real-world tasks require decisions at varying granularities, and humans excel at this by leveraging a unified cognitive representation where planning is fundamentally understood as a high-level form of action. However, current Large Language Model (LLM)-based agents lack this crucial capability to operate fluidly across decision granularities. This limitation stems from existing paradigms that enforce a rigid separation between high-level planning and low-level action, which impairs dynamic adaptability and limits generalization. We propose **ReCode** (**Re**cursive **Code** Generation), a novel paradigm that addresses this limitation by unifying planning and action within a single code representation. In this representation, ReCode treats high-level plans as abstract placeholder functions, which the agent then recursively decomposes into finer-grained sub-functions until reaching primitive actions. This recursive approach dissolves the rigid boundary between plan and action, enabling the agent to dynamically control its decision granularity. Furthermore, the recursive structure inherently generates rich, multi-granularity training data, enabling models to learn hierarchical decision-making processes. Extensive experiments show ReCode significantly surpasses advanced baselines in inference performance and demonstrates exceptional data efficiency in training, validating our core insight that unifying planning and action through recursive code generation is a powerful and effective approach to achieving universal granularity control.

## 1 Introduction

Human decision-making in the real world naturally operates across varying granularities, seamlessly integrating high-level planning with fine-grained actions. Consider the simple act of preparing breakfast: one effortlessly shifts from deciding a high-level plan like "making bacon and eggs" to executing detailed, motor-level actions such as "cracking an egg". Cognitive science attributes this fluid adaptability to the human brain's integrated approach to control, which allows for decisions to be managed across different granularities. This integration is reflected in the brain's shared representations for related cognitive processes (Prinz, 1997) and its hierarchical structure (Koechlin et al., 2003; Badre & D'esposito, 2009). Thus, humans naturally master this dynamic control of decision granularity, an essential capability for adaptive, intelligent behavior.

Achieving this adaptive intelligence is a primary goal for LLM-based agents (Liu et al., 2025), yet current frameworks fall short because they operate at fixed granularities. The core issue is that existing approaches explicitly separate planning from action into distinct decision-making processes. For example, ReAct (Yao et al., 2023) agent as shown in Figure 1(a), alternates strictly between reasoning and primitive actions step-by-step, limiting it to fine-grained decision-making without strategic foresight. On the other hand, agent with planner module (Figure 1(b)) separates high-level planning from low-level action using predefined structures. Such rigid boundaries impede agents from dynamically adjusting their decision granularity in response to evolving task complexities, ultimately resulting in brittle performance in complex, real-world environments.

To address this failure, our key insight is that planning and action are not fundamentally distinct cognitive processes, but rather represent decisions at different levels of granularity. At its essence, a plan is simply a higher-level action, analogous to how pseudo code represents higher-level conceptual thinking compared to concrete, executable code. Motivated by this insight, we propose

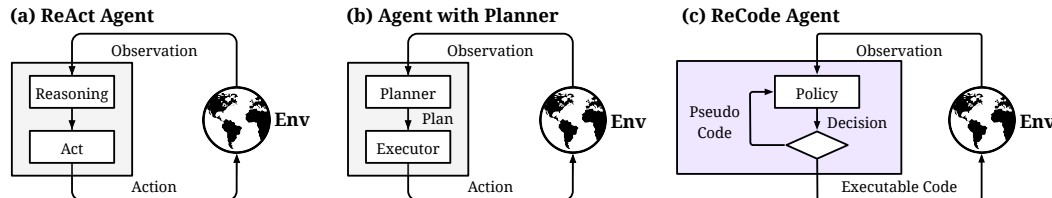

Figure 1: A comparison of LLM-based agent decision-making paradigms. **(a) ReAct Agent** operates in a simple observation-action loop at a fixed, fine-grained level. **(b) Agent with Planner** enforces a rigid separation between a high-level Planner and a low-level Executor, limiting adaptability. **(c) ReCode Agent** unifies plan and action in a code representation. The policy recursively refines high-level plans until primitive actions within a single dynamic loop, enabling fluid control over decision granularities.

**RECODE** (**Re**cursive **Code** Generation), a novel paradigm that applies this idea by unifying plan and action within a single code representation. As illustrated in Figure 1(c), ReCode implements this pseudo-code-like concept by treating high-level plans as abstract placeholder functions, which the agent then recursively decomposes into finer-grained sub-functions until reaching executable primitive actions. This recursive process inherently dissolves the rigid boundary between planning and action, enabling the agent to dynamically control its decision granularity.

Furthermore, ReCode's recursive structure provides a significant training advantage by inherently generating rich, hierarchical data. Unlike traditional approaches that yield flat action sequences, ReCode naturally produces structured decision trees capturing the entire cognitive process, from high-level planning down to executable actions. This comprehensive, multi-granularity training data significantly enhances the agent's ability to learn complex task decompositions and adaptive decision-making strategies, improving both generalization and data efficiency.

Extensive experiments conducted across diverse, complex decision-making benchmarks validate the effectiveness of the ReCode paradigm. ReCode consistently surpasses strong paradigm baselines such as ReAct (Yao et al., 2023) and CodeAct (Wang et al., 2024b), delivering substantial improvements in overall task performance and problem-solving efficiency. Furthermore, our experiments reveal that the hierarchical, multi-granular data generated by ReCode plays a critical role in training flexible decision-making capabilities. These empirical results demonstrate ReCode's potential as a transformative approach to building more adaptive, capable, and generalizable LLM-based agents.

In summary, our contributions are as follows:

- We identify the lack of flexible decision granularity control as a fundamental limitation in existing LLM-based agents, which stems from the rigid separation of plan and action.

- We propose ReCode, a novel agent paradigm that achieves universal granularity control by unifying plan and action within a single code representation. This unification is realized through a recursive mechanism where the agent decomposes high-level placeholder functions until primitive actions.

- We demonstrate that this recursive structure inherently generates hierarchical, multi-granularity data, which enables models to learn hierarchical reasoning processes and leads to significant improvements in training data efficiency.

- We validate our approach through comprehensive experiments, showing that ReCode significantly surpasses strong baselines in both inference performance and training efficiency across diverse benchmarks.

## 2 RELATED WORK

The powerful reasoning and language capabilities of Large Language Models (LLMs) have inspired significant research effort to create agents that can interact with complex environments and accomplish specific tasks (Ahn et al., 2022; Huang et al., 2023; Yao et al., 2023). These efforts have largely fallen into two dominant paradigms: agents following the ReAct paradigm (Yao et al., 2023)

and agents with explicit planners. However, both approaches, in their current forms, struggle with the fundamental limitation of a rigid separation between high-level planning and low-level action, which prevents flexible decision granularity control.

**LLM-based ReAct Agent** The ReAct paradigm (Yao et al., 2023) represents a foundational approach in LLM-based agents, interleaving reasoning and primitive actions in a step-by-step loop. Building on this foundation, code has emerged as a more expressive medium for action specification, with paradigms like CodeAct (Wang et al., 2024b; Ni et al., 2025) leveraging the extensive code pre-training capabilities of LLMs. These foundational paradigms have inspired a rich ecosystem of agent systems (Yang et al., 2023a; Wu et al., 2024; Zhang et al., 2025; Liang et al., 2025; Hu et al., 2025). Subsequent research has primarily focused on optimizing the step-by-step execution loop through two main directions, including synthetic data generation methods (Zelikman et al., 2022; Yang et al., 2024b; Wang et al., 2024a) and novel training algorithms (Song et al., 2024; Qiao et al., 2024; Du et al., 2024; Lee et al., 2024; Xia et al., 2025; Xiong et al., 2025; Cao et al., 2025). While these optimization efforts yield performance improvements, they maintain the paradigm's fundamental characteristic of lacking high-level planning. This limitation restricts the agent's foresight in complex, long-horizon tasks, where reasoning about future states and multi-step consequences becomes critical.

**LLM-based Agent Planning** Planning emerged as a distinct research thrust, specifically to address this lack of foresight in ReAct Agent paradigm. Early approaches often adopted a "plan-and-execute" strategy, where a complete natural language plan is generated upfront before any actions are taken (Wang et al., 2023a; Yang et al., 2023b; Kagaya et al., 2024; Sun et al., 2024a). However, these static plans are often brittle and difficult to adapt to dynamic environments. To address this, more sophisticated methods emerged, such as hierarchical planning frameworks (Paranjape et al., 2023; Sun et al., 2024b), dynamic re-planning (Sun et al., 2023; Prasad et al., 2024) and meta-control (Yuan et al., 2025) methods, which continuously modify their plans based on environmental feedback. In addition, some recent work (Shentu et al., 2024) attempts to relax the limitations of natural-language plans as the sole interface between a high-level planner and a low-level controller by introducing learned latent interfaces. Nonetheless, these traditional planning methods still share the fundamental flaw of maintaining a rigid separation between the high-level plan and the low-level action execution.

**LLM-based Agent with Recursive Structure** While some recent approaches have begun to bridge this gap by explicitly integrating recursion or code-based processes, they still do not achieve a full unification and adaptive granularity control. Code-as-Policies (Liang et al., 2023) and follow-up methods (Wang et al., 2023b; Mu et al., 2024; Ahn et al., 2025) use complete executable programs to represent the agent policy, and recursively expand high-level sketches into full programs to allocate context across different sub-tasks, but lack mechanisms to dynamically adjust the plan based on interaction with the environment. THREAD and REPL-Plan (Schroeder et al., 2025; Liu et al., 2024) instead introduce recursive substructures on top of ReAct-style agents, descending into sub-tasks to isolate local context in complex problems, but this recursion is primarily used for internal context management and the overall interaction pattern remains a standard step-by-step action loop.

It is highlighted that the core limitation across all existing paradigms remains this rigid separation of plan and action. In contrast, our work is built on the key insight that plan and action are not distinct processes, but rather represent decisions at different levels of granularity. This forms the foundation for our paradigm, which is unifying plan and action in a single representation.

## 3 METHODOLOGY

### 3.1 PRELIMINARY

**LLM-based Agent Decision Process** We model the interaction between an LLM-based agent and its environment as a simplified decision-making process $\mathcal{M} = \langle \mathcal{S}, \mathcal{A}, \mathcal{O}, T, R \rangle$, where $\mathcal{S}$ is the state space, $\mathcal{A}$ is the primitive action space, $\mathcal{O}$ is the observation space, $T : \mathcal{S} \times \mathcal{A} \rightarrow \mathcal{S}$ is the transition function, and $R : \mathcal{S} \times \mathcal{A} \rightarrow \mathbb{R}$ is the reward function. At each step, the agent receives an observation $o \in \mathcal{O}$ and produces decisions that ultimately translate into executable primitive actions $a \in \mathcal{A}$.

Beyond the primitive action space, we introduce the plan space $\mathcal{P}$, which encompasses intentions and goals at coarser granularities that cannot be directly executed but must be refined into sequences of

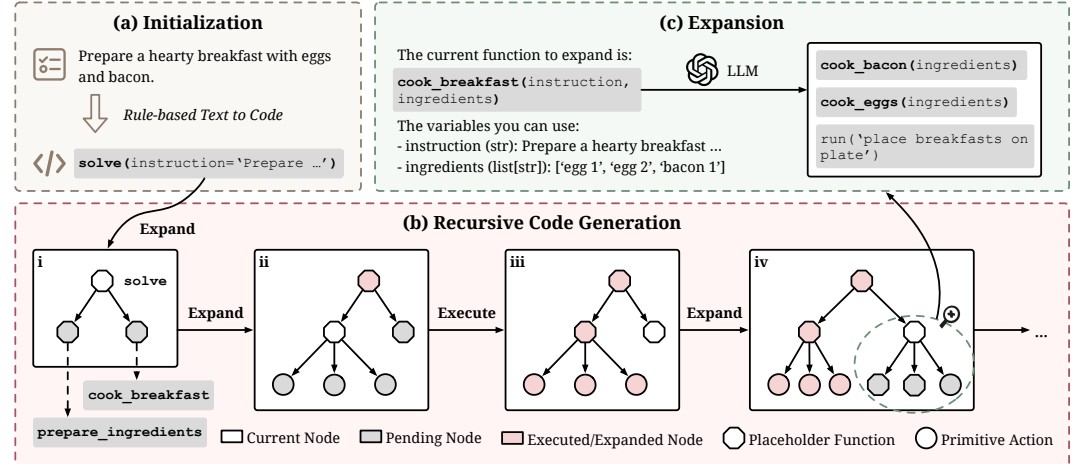

Figure 2: An overview of the ReCode. (a) The task instruction is transformed into an initial place-holder function via a rule-based text-to-code method. (b) The system traverses the tree depth-first, automatically executing the code of current node and expanding placeholder functions into child nodes when encountered. (c) LLM-based expansion operates with clean context. Only the current function signature and available variables are provided, without any tree structure or execution history.

primitive actions or intermediate sub-plans. We define the decision space as $\mathcal{D} = \mathcal{A} \cup \mathcal{P}$, representing all possible outputs an agent can generate across different granularities.

Current agent paradigms typically operate with predefined, fixed decision spaces. ReAct agents directly select from a pre-determined action set $\mathcal{A}$, while planner-based agents generate sequences over the same fixed $\mathcal{A}$ or rely on manually specified plan templates. This constraint fundamentally limits adaptability, as agents cannot dynamically generate novel decisions suited to unforeseen contexts.

**Decision Granularity**   Real-world tasks demand decisions at varying granularities. Fine-grained decisions in $\mathcal{A}$ correspond to immediately executable primitive actions such as run('crack egg'), while coarse-grained decisions in $\mathcal{P}$ represent higher-level intentions requiring decomposition, such as prepare_breakfast().

The granularity forms a natural hierarchy. Consider the breakfast preparation example. The decision "prepare breakfast" is coarser than "cook eggs", which is in turn coarser than "crack egg". Each level encompasses broader objectives and longer temporal horizons. The finest level consists of executable primitive actions, while the coarsest level represents the complete task specification.

Motivated by the perspective that plans are essentially high-level actions at different abstraction levels, ReCode unifies decisions across all granularities within a single code representation. High-level plans in $\mathcal{P}$ take the form of placeholder functions, which are recursively refined into finer-grained components until reaching executable primitive actions in $\mathcal{A}$. This formulation enables the agent to dynamically generate decisions, whether plans or actions, at appropriate granularities for the given context, effectively creating an unbounded decision space.

### 3.2 METHOD OVERVIEW

We introduce ReCode, an LLM-based agent paradigm with recursive code generation that achieves universal control of decision granularity. This paradigm begins by unifying plan and action into the same representation, enabling the entire decision-making process to be implemented through a single, consistent operation. This unification naturally gives rise to a tree structure where the highest-level task can be recursively decomposed until executable leaves. This dynamic process of building and executing a decision tree is illustrated in Figure 2.

**Unify Plan and Action**   The key insight of ReCode is that plans and actions, despite their apparent differences, can be unified under a single executable code representation. This unification addresses

a fundamental limitation in current LLM-based agent frameworks: the rigid separation between abstract planning and concrete primitive action, which prevents flexible decision granularity control.

We represent both plans and actions as Python function calls. This establishes a common computational substrate for the agent's policy. Actions are executable and environment-specified that can directly interface with the environment, such as `run('click the submit button')`. Plans are expressed as unimplemented placeholder functions. These functions encapsulate sub-goals that require further expansion, like `prepare_breakfast()` and `get_ingredients()`. For example, a high-level plan `prepare_breakfast()` can be expanded by the policy into a code block containing other plans, like `get_ingredients()` and `cook_meal()`. These, in turn, are recursively broken down until they resolve into a sequence of primitive actions like `run('open refrigerator')` and `run('turn on stove')`.

**Recursive Code Generation** Building on this unified representation, ReCode operates via a recursive generation and execution loop, as detailed in Algorithm 1. The process begins with a rule-based text-to-code transformation that converts the task instruction and initial observation into a placeholder function, which is the root node. The agent's policy model $\pi$ then expands the current node into a child code block.

An executor processes the generated code block sequentially. When encountering a primitive action, the executor directly executes it in the environment. When encountering a placeholder function, the executor triggers an expansion by invoking ReCode again with this placeholder as the new current node. The decision tree grows in this process, gradually breaking down the task into more detailed plans until all leaves are executable primitive actions.

The recursion terminates when a generated code block contains only primitive actions. Execution then returns to the previous, coarser level. This process continues until the initial top-level placeholder is fully resolved, allowing the agent to seamlessly navigate across granularity levels. The policy adaptively determines when to maintain abstract planning and when to commit to concrete actions based solely on the current decision context, without explicit granularity supervision. The resulting execution trace forms a hierarchical decision tree that captures the complete reasoning process from strategic planning to primitive execution.

---

**Algorithm 1** The ReCode Algorithm

---

**Require:** Task $T$, Policy $\pi$, Environment $E$, current node $c$
1: **procedure** RECODE($T, \pi, E, c$)
2:     **if** $c$ is NONE **then**                                       ▷ Initialization
3:         $o_0 \leftarrow$ RESET($E$)         ▷ Reset environment and get initial observation
4:         $c \leftarrow$ TEXT2CODE($T, o_0$)         ▷ Convert task to root placeholder
5:     **end if**
6:     code block $\leftarrow \pi(c)$         ▷ Generate children for current placeholder
7:     **for** each child code unit $u$ in code block **do**
8:         **if** ISPRIMITIVE($u$) **then**                           ▷ Primitive action
9:             EXECUTE($u, E$)             ▷ Execute in environment
10:        **else**                                 ▷ Placeholder function
11:            RECODE($T, \pi, E, u$)          ▷ Recursive expansion
12:         **end if**
13:     **end for**
14: **end procedure**

---

## 3.3 IMPLEMENTATION DETAILS

Bridging the ReCode paradigm from theory to practice requires several key engineering efforts to ensure robust and stable agent performance.

**Task Initialization** The starting point for ReCode is converting the natural language task instruction into the root placeholder function. We employ a simple rule-based method that directly encapsulates the task instruction and initial observation as string arguments within a predefined template, such as `solve(instruction, observation)`. This task-agnostic approach delegates the full responsibility of interpreting and decomposing the task to the learned policy model, compelling it to develop

core planning capabilities from raw inputs and enhancing generalization across diverse tasks and environments.

**Context Management** ReCode implements context management through a unified variable namespace that persists throughout task execution. When expanding a placeholder function, the system serializes all currently provided variables and their values into a structured text description, which is injected into the prompt as context. After executing the generated code block, newly created or modified variables are updated in this shared namespace. This design creates a hierarchical information flow where sub-tasks can access context established by parent tasks at any level of the call stack. Crucially, the policy model only sees the current variable state, not the full execution history. This enforces explicit state management. The model must learn to consciously save important information to variables for future use. For example, capturing an action's output as `obs = run('go to cabinet 1')` allows subsequent inspection of the `obs` variable. This approach maintains concise and relevant context while guiding the model toward structured and deliberate planning.

**Error Handling and Recursion Control** To ensure robust execution, ReCode addresses two practical challenges. First, code generated by LLMs is susceptible to syntax or runtime errors. We employ a self-correction loop during inference. Upon execution failure, the system re-invokes the policy with the original placeholder and the error traceback as additional context, allowing it to recover from transient generation failures within a single trajectory. Second, unbounded recursion can lead to infinite loops or excessively deep decomposition trees. We impose a maximum recursion depth of 10 for our benchmark tasks, chosen as a conservative upper bound above the empirically optimal depth (in Section 4.2), balancing planning complexity with guaranteed termination.

## 4 EXPERIMENT

### 4.1 EXPERIMENTAL SETUP

**Environments** We conduct experiments across three text-simulated environments that represent diverse decision-making challenges for LLM-based agents. ALFWorld (Shridhar et al., 2021) provides an embodied household environment where agents navigate and manipulate objects to complete daily tasks. WebShop (Yao et al., 2022) simulates an online shopping scenario requiring agents to search, compare, and purchase products based on specific requirements. ScienceWorld (Wang et al., 2022) presents a scientific laboratory setting where agents conduct experiments and manipulate scientific instruments. All three environments are partially observable Markov decision processes where agents cannot fully infer the environment state from observations alone. For more detailed descriptions of the environments, please refer to Appendix A.1.

**Baselines** To evaluate the effectiveness of ReCode, we divide our experiments into the inference part and the training part.

For inference experiments, we compare against several mainstream paradigms: ReAct (Yao et al., 2023), which alternates between reasoning and action for iterative environment interaction; CodeAct (Wang et al., 2024b), extending ReAct's action space from natural language to Python code. And some of the work focused on improving LLM-based agent planning: AdaPlanner (Sun et al., 2023), which pre-writes action sequences in Python and iteratively modifies them based on environmental feedback; and ADaPT (Prasad et al., 2024), which decomposes tasks on-demand using ReAct as the sub-task executor.

For training experiments, which evaluate how well a paradigm's data structure lends itself to learning, we conduct supervised fine-tuning (SFT) (Wei et al., 2022) experiments on Re-Code, ReAct, and Code-Act to directly compare their learning efficiency. Also, we compare some advanced training methods on

| Method | Metric | ALFWorld | ScienceWorld | WebShop |
|--------|--------|----------|--------------|---------|
| ReAct | Data Pairs | 27,607 | 12,833 | 7,181 |
| | Tokens | 18,958,782 | 15,356,801 | 9,409,144 |
| | Avg. Reward% ($\pm$Std) | 100.0* | 100.0 $\pm$ 0.00 | 82.3 $\pm$ 21.15 |
| CodeAct | Data Pairs | 5,049 | 5,984 | 3,853 |
| | Tokens | 9,630,822 | 15,189,063 | 9,214,185 |
| | Avg. Reward% ($\pm$Std) | 100.0* | 60.8 $\pm$ 24.51 | 94.6 $\pm$ 10.36 |
| ReCode | Data Pairs | 6,385 | 3,500 | 2,473 |
| | Tokens | 4,962,898 | 3,986,769 | 2,662,276 |
| | Avg. Reward% ($\pm$Std) | 100.0* | 88.5 $\pm$ 14.58 | 97.5 $\pm$ 7.05 |

Table 1: Statistics of the SFT datasets. *For ALFWorld, all filtered trajectories are successful tasks with a reward of 1.

the ReAct-style agent, including ETO (Song et al., 2024), which learns from expert trajectories via Direct Preference Optimization (Rafailov et al., 2023); and WKM (Qiao et al., 2024), introducing parameterized world models for training on expert and synthetic trajectories. Due to differences in models, training setups, or environment configurations, we could not directly implement these latter works, but list them for reference.

**Evaluation**  We report average reward (%) as our primary metric across all experiments. The environments employ different reward settings. ALFWorld provides binary rewards (0 or 1) indicating task completion, while WebShop and ScienceWorld offer dense rewards ranging from 0 to 1 based on task progress and accuracy. ALFWorld and ScienceWorld both include out-of-distribution evaluation sets to assess generalization capabilities.

**Implementation Details**  For inference experiments, all methods use GPT-4o mini (Hurst et al., 2024) to ensure fair comparison. We adapt few-shot prompts from prior work, making minimal modifications to fit each method's format while keeping the number of examples consistent. The few-shot examples used for ReCode are provided in Appendix A.3.

For training experiments, we use Qwen2.5-7B-Instruct (Yang et al., 2024a) as the base model. We construct the training datasets through the following process: First, for each training instance, we use a powerful teacher model, DeepSeek-V3.1 (DeepSeek-AI, 2024), to generate trajectories using ReAct, CodeAct, and ReCode respectively. Second, we filter these trajectories by keeping only the top 40% based on final reward. Third, from each retained trajectory, we extract input-output pairs as SFT training data, where the input excludes few-shot examples to focus on the model's learning of the paradigm itself.

Table 1 presents detailed statistics of the resulting training datasets. As shown, different paradigms produce vastly different amounts of training data from the same pool of source instances. ReCode generates significantly fewer but more compact data pairs (e.g., 6,385 pairs with 4.96M tokens on ALFWorld) compared to ReAct (27,607 pairs with 18.96M tokens). All models are trained using full-parameter fine-tuning on 8 NVIDIA A800 80GB GPUs with the open-source verl framework (Sheng et al., 2024).

## 4.2 INFERENCE RESULTS

| Method | ALFWorld | | ScienceWorld | | WebShop | Average |
|---|---|---|---|---|---|---|
| | Seen | Unseen | Seen | Unseen | | |
| ReAct   GPT-4o mini | 59.29 | 64.18 | **47.02** | 38.91 | 27.62 | 47.4 |
| CodeAct   GPT-4o mini | 72.14 | 85.07 | 22.68 | 18.41 | 21.37 | 43.9 |
| AdaPlanner   GPT-4o mini | 75.00 | 92.54 | 28.49 | 26.11 | 29.17 | 50.3 |
| ADaPT   GPT-4o mini | 34.29 | 41.04 | 37.53 | 32.94 | 32.79 | 35.7 |
| ReCode   GPT-4o mini | **83.57** | **96.27** | 42.78 | **41.18** | **39.97** | **60.8** |
| ReAct   Gemini 2.5 Flash | 85.71 | 85.07 | 24.73 | 25.60 | 39.74 | 52.2 |
| CodeAct   Gemini 2.5 Flash | 52.86 | 63.43 | 23.89 | 20.64 | 43.67 | 40.9 |
| AdaPlanner   Gemini 2.5 Flash | 77.86 | 85.82 | 30.41 | 33.25 | 8.68 | 47.2 |
| ADaPT   Gemini 2.5 Flash | 64.29 | 73.88 | 37.81 | 31.44 | 33.95 | 48.3 |
| ReCode   Gemini 2.5 Flash | **90.00** | **96.27** | **54.35** | **43.77** | 46.57 | **66.2** |
| ReAct   DeepSeek-V3.1 | 90.71 | 88.06 | **68.52** | **61.75** | 22.94 | 66.4 |
| CodeAct   DeepSeek-V3.1 | 51.43 | 59.70 | 35.96 | 27.91 | 36.50 | 42.3 |
| AdaPlanner   DeepSeek-V3.1 | 87.86 | 92.54 | 27.75 | 26.25 | 36.65 | 54.2 |
| ADaPT   DeepSeek-V3.1 | 32.86 | 32.09 | 35.40 | 34.81 | 31.93 | 33.4 |
| ReCode   DeepSeek-V3.1 | **95.71** | **94.78** | 50.20 | 50.21 | **55.07** | **69.2** |

Table 2: Average rewards (%) on three environments in the few-shot inference setting. The table is divided into three sections based on the underlying model: GPT-4o mini, Gemini 2.5 Flash, and DeepSeek-V3.1. The optimal and suboptimal results in each section are marked in **bold** and underlined, respectively.

**Main Results**  As shown in Table 2, ReCode achieved significant performance improvements across all three environments, with an average score of 60.8, surpassing the best baseline method by approximately 10.5 (relative 20.9%). Notably, compared to the ReAct and CodeAct paradigms,

ReCode improved average performance by 13.4 and 16.9, respectively. The results demonstrate ReCode's exceptional capabilities across diverse task domains. In the ALFWorld environment, our method exhibits particularly strong performance, achieving a high score of 83.57 on seen tasks and an outstanding 96.27 on unseen tasks, substantially outperforming all baseline methods. On WebShop, ReCode maintains competitive performance with a score of 39.97, representing a 21.9% improvement over ADaPT. In the ScienceWorld environment, while ReCode shows comparable performance to ReAct on seen data, it demonstrates notable improvement on unseen tasks. These consistent improvements across varied environments underscore the effectiveness of our approach in enhancing both task-specific performance and cross-domain adaptability. We provide detailed case study in Appendix A.5, showcasing representative trajectory from ALFWorld to illustrate ReCode's decision-making process.

**Multiple Model Results** To illustrate that the ReCode paradigm is applicable to most models, we also experiment with Gemini Flash 2.5 (Comanici et al., 2025) and DeepSeek-V3.1 (DeepSeek-AI, 2024), with results shown in Table 2. The cross-model evaluation demonstrates ReCode's robustness and generalizability beyond a single language model architecture. On Gemini 2.5 Flash, ReCode achieved an average performance of 66.2, maintaining consistent improvements over baseline methods. Similarly, when applied to DeepSeek-V3.1, our approach obtained a 69.2 average score, further validating the paradigm's effectiveness. These results indicate that ReCode's performance gains are not dependent on specific model characteristics or training procedures, but rather stem from the fundamental improvements in the reasoning and code generation process.

**Recursion Depth Analysis** To validate our depth control mechanism, we conduct ablation experiments with varying maximum depth limits on the ScienceWorld seen set using GPT-4o mini. As shown in Figure 3, performance exhibits a clear inverted-U pattern, peaking at depth 8 (43.22%). Shallow depths restrict hierarchical decomposition, while excessive depths decline performance, likely due to over-decomposition fragmenting the decision-making process. The optimal range lies between depths 6-12. Based on these findings, we set the maximum depth to 10 across all environments, providing a conservative upper bound that prevents unbounded recursion while rarely constraining practical reasoning.

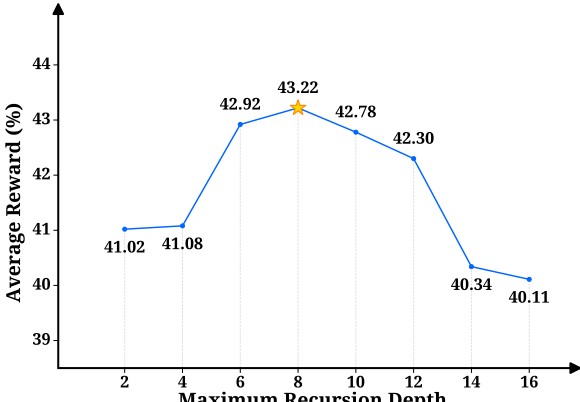

Figure 3: Effect of maximum recursion depth on agent performance in ScienceWorld seen set using GPT-4o mini. The star indicates the optimal depth achieving peak performance.

**Cost Analysis** As shown in Table 3, ReCode is remarkably cost-efficient. On average, a ReCode trajectory costs 78.9% less than ReAct and 84.4% less than CodeAct. This dramatic improvement stems from ReCode's structured exploration. Instead of the extensive, token-intensive, step-by-step reasoning cycles characteristic of reactive methods, ReCode's hierarchical decomposition allows the agent to make more abstract and potent decisions. This leads to shorter, more direct reasoning paths, significantly reducing the total tokens required to solve a task while achieving superior performance.

| Method | ALFWorld | | ScienceWorld | | WebShop | Average |
| --- | --- | --- | --- | --- | --- | --- |
| | Seen | Unseen | Seen | Unseen | | |
| ReAct | 10.55 | 9.76 | 10.29 | 11.07 | 3.15 | 8.96 |
| CodeAct | 5.24 | 5.92 | 13.42 | 12.13 | 24.04 | 12.15 |
| ReCode | 2.11 | 1.99 | 1.91 | 2.23 | 1.19 | 1.89 |

Table 3: The average cost ($\times 10^{-3}$ US dollars) on all benchmark environments using GPT-4o mini. All cost calculations refer to the official API pricing for GPT-4o mini.

| Method | ALFWorld | | ScienceWorld | | WebShop | Average |
|---|---|---|---|---|---|---|
| | Seen | Unseen | Seen | Unseen | | |
| ReAct   Qwen2.5-7B-Instruct | 57.86 | 62.69 | 19.49 | 14.85 | 32.33 | 37.4 |
| CodeAct   Qwen2.5-7B-Instruct | 67.86 | 69.40 | 13.47 | 9.29 | 37.43 | 39.5 |
| ReAct+SFT   Qwen2.5-7B-Instruct | 90.00 | 90.30 | **60.28** | **54.64** | 42.61 | 67.6 |
| CodeAct+SFT   Qwen2.5-7B-Instruct | 86.43 | 88.06 | 37.25 | 33.59 | 33.48 | 55.8 |
| ReAct+ETO* Llama-3-8B-Instruct | 64.29 | 64.18 | 57.90 | 52.33 | 64.57 | 60.7 |
| ReAct+WKM* Llama-3-8B-Instruct | 68.57 | 65.93 | 60.12 | 54.75 | 66.64 | 63.2 |
| ReCode   Qwen2.5-7B-Instruct | 66.43 | 73.88 | 15.84 | 15.38 | 42.24 | 42.8 |
| ReCode+SFT   Qwen2.5-7B-Instruct | **92.14** | **97.01** | 59.85 | 52.39 | **50.85** | **70.4** |

Table 4: Average rewards on three environments in the training setting. The optimal and suboptimal results in each section are marked in **bold** and underlined, respectively. ($^*$) denotes the methods without our implementation, which were reported in the original paper of WKM.

## 4.3 TRAINING RESULTS

**Main Results**   Table 4 presents the training results using Qwen2.5-7B-Instruct. ReCode+SFT achieves a strong average performance of 70.4% across all environments, surpassing both ReAct+SFT (67.6%) and CodeAct+SFT (55.8%).

Notably, this performance is achieved with remarkable data efficiency. As shown in Table 1, ReCode's training set contains only 3,500 pairs, 3.7 times less than the 12,833 pairs used by ReAct, and its average reward (88.5%) is also lower than ReAct's (100.0%). This demonstrates that ReCode+SFT can achieve superior results even with significantly less and potentially lower-quality training data. On individual benchmarks, ReCode+SFT shows dominant performance on ALFWorld and WebShop, and remains competitive on ScienceWorld.

**Data Efficiency Analysis**   To systematically investigate the source of ReCode's superior data efficiency, we conducted training on data subsets filtered by reward percentile $p \in \{10, 20, 30, 40\}$. Figure 4 and Table 5 present the results on ScienceWorld. The results reveal distinct scaling behaviors. ReCode achieves 55.34% on seen tasks at $p = 20$ using only 1,713 pairs, improving to 59.85% at $p = 40$ with 3,500 pairs. In contrast, ReAct requires 12,833 pairs at $p = 40$ to reach 60.28% on seen tasks. At matched percentiles, ReCode consistently outperforms ReAct with substantially fewer data pairs. At $p = 10$, ReCode achieves 44.87% with 688 pairs versus ReAct's 34.05% with 3,094 pairs (4.5× more). At $p = 20$, ReCode reaches 55.34% with 1,713 pairs versus ReAct's 36.34% with 6,320 pairs (3.7× more). Figure 4 shows that ReCode's curves rise steeply then plateau, while ReAct's curves show gradual improvement requiring substantially more data, confirming that ReCode's hierarchical structure provides richer learning signals per training example.

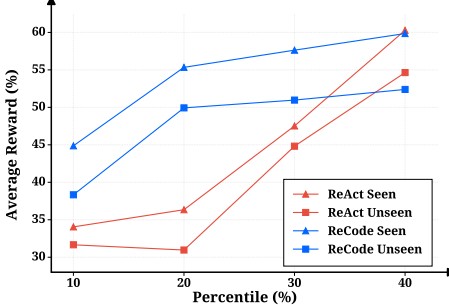

| Method | Percentile | Data Pairs | ScienceWorld | |
|---|---|---|---|---|
| | | | Seen | Unseen |
| ReAct | 10 | 3,094 | 34.05 | 31.66 |
| | 20 | 6,320 | 36.34 | 30.96 |
| | 30 | 9,488 | 47.52 | 44.82 |
| | 40 | 12,833 | 60.28 | 54.64 |
| ReCode | 10 | 688 | 44.87 | 38.33 |
| | 20 | 1,713 | 55.34 | 49.93 |
| | 30 | 2,499 | 57.63 | 50.97 |
| | 40 | 3,500 | 59.85 | 52.39 |

Figure 4: Performance of ReCode (blue) and ReAct (red) on ScienceWorld under different filtering percentiles. Performance on seen tasks is represented by triangles, and performance on unseen tasks is represented by squares.

Table 5: Training data statistics and corresponding ScienceWorld performance for ReAct and ReCode. The table lists the number of data pairs and the resulting average rewards (%) for each method at different filtering percentiles ($p$).

## 4.4 EVALUATION ON EMBODIED MANIPULATION TASKS

To further evaluate whether ReCode's unified granularity control benefits tasks that require both high level planning and low level motor control, we conduct experiments on the ManiSkill (Tao et al., 2025) benchmark. Unlike ALFWorld, WebShop, and ScienceWorld, ManiSkill requires continuous control of robot end effectors and fine grained pose adjustments, which provides a natural setting where dynamic adjustment of decision granularity is essential. We use two standard manipulation tasks, *PickCube v1* and *PullCube v1*, based on the official ManiSkill3 Franka Panda robot. ReCode interacts with the environment through a small set of textual actions with continuous parameters; details of the wrappers, low level APIs, and task specifications are provided in Appendix A.1.

We compare ReCode with two widely used paradigms for embodied decision making, ReAct and Code-as-Policies (CaP) (Liang et al., 2023), where ReAct performs step by step reasoning and emits actions, and CaP represents policies as full programs and recursively fills in missing functions before execution. We evaluate all methods with two LLMs, GPT-4o mini and Gemini 2.5 Flash, and report the success rate (%) on 50 episodes with random initialization under fixed seed for each task. The results in Table 6 show that ReAct struggles on Man-

| Model | Method | PickCube | PullCube |
|-------|--------|----------|----------|
| GPT-4o mini | ReAct | 0 | 2 |
| | CaP | 34 | 34 |
| | ReCode | 78 | 46 |
| Gemini 2.5 Flash | ReAct | 40 | 30 |
| | CaP | 82 | 44 |
| | ReCode | 90 | 100 |

Table 6: ReAct, Code-as-Policies (CaP), and Re-Code's success rates (%) on the two tasks in Man-iSkill: *PickCube v1* and *PullCube v1*.

iSkill, CaP improves performance but still relies on static programs that cannot adapt during execution, while ReCode consistently achieves the best results. The experiment indicate that ReCode's unified recursive code based framework transfers to embodied manipulation and that flexible control of decision granularity yields practical benefits beyond text only environments.

## 5 CONCLUSION

In this work, we introduced ReCode, a novel paradigm for LLM-based agents that achieves universal control of decision granularity through recursive code generation. By unifying plan and action within a single code representation, ReCode addresses fundamental limitations in current agent paradigms that rigidly separate decisions of different granularity. Our key insight is that planning is fundamentally high-level action at different granularities. This enables agents to dynamically decompose goals into hierarchical code structures without reliance on predefined action spaces. Extensive experiments across three diverse environments demonstrate ReCode's dual advantages. In inference, ReCode achieves superior performance with over 20.9% improvement. In training, ReCode exhibits remarkable efficiency using dramatically fewer data while achieving better results. The recursive structure naturally generates rich, multi-granularity training data that captures complete cognitive processes from strategic planning to concrete action. ReCode establishes a foundation for more adaptive AI agents that can fluidly transition across decision granularities based on task complexity and situational demands.

## 6 LIMITATIONS AND FUTURE WORK

While ReCode shows strong potential, its effectiveness still depends heavily on the underlying language model and the quality of the few shot examples. The framework requires the model to produce structured code in a specific format, so weaknesses in planning ability or deviations from the required format can lead to unstable performance. A natural direction for future work is to strengthen the model's understanding of the ReCode framework through targeted pre training or fine tuning, and to explore reinforcement learning signals that reward efficient hierarchical plans and reliable expansions. It is also promising to develop mechanisms that increase robustness to generation errors and to investigate curriculum style training procedures that gradually introduce more complex tasks and reasoning patterns.

## ETHICS STATEMENT

We have read and will adhere to the ICLR Code of Ethics and the ICLR Code of Conduct. Our research introduces ReCode, a framework for LLM-powered agents, and evaluates it within simulated environments. The datasets used in our study are well-established public benchmarks for academic research. These environments do not contain any personally identifiable information (PII) or sensitive real-world data. Our work did not involve human subjects, crowd-sourcing, or the scraping of private data; therefore, Institutional Review Board (IRB) approval was not required.

We acknowledge that research on autonomous agents carries potential dual-use risks. To mitigate these, our experiments are intentionally confined to benign, closed-world tasks such as online shopping and household activities within simulated settings. We followed good scholarly practice by reporting our methods and results transparently and citing prior work accurately. The authors declare no competing interests or external sponsorships that could have influenced the outcomes of this research.

## REPRODUCIBILITY STATEMENT

We are committed to ensuring the reproducibility of our research. All essential details for reproducing our results are provided within this paper. The descriptions of the datasets (ALFWorld, WebShop, ScienceWorld) and their respective splits are detailed in the Appendix. Our experimental setup, including the specific models used for inference and fine-tuning, training configurations, and evaluation protocols, is described in Section 4. To facilitate full replication, we will release our code and few-shot prompts as supplementary material. An anonymous repository containing these artifacts is available at: `https://anonymous.4open.science/r/ReCode-ICLR26-E7E4`.

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

# A APPENDIX

## A.1 DATASETS

**ALFWorld**  ALFWorld (Shridhar et al., 2021) provides a text-only game environment where the agent is required to complete a series of long-horizon household tasks based on natural language instructions. Built upon the embodied vision benchmark ALFRED (Shridhar et al., 2020), ALFWorld inherits its six core task types: pick and place, examine in light, clean and place, heat and place, cool and place, and pick two and place. After executing a command (e.g., "go to countertop 1"), the agent receives immediate textual feedback from the environment (e.g., "You arrive at countertop 1. On the countertop 1, you see an egg 3, a peppershaker 1, and a tomato 1"). ALFWorld only provides a binary reward (1/0) to indicate whether the task was successfully completed.

**WebShop**  WebShop Yao et al. (2022) is an online shopping website environment constructed with 1.18 million real-world product data points. The agent must navigate the site by searching and clicking buttons to browse various products, with the goal of finding and purchasing a specific item that meets the given requirements. We follow the data partitioning from MPO (Xiong et al., 2025), using a test set of 200 instances. WebShop provides a dense reward between 0 and 1 based on the agent's actions and interactions with products. A task is considered successful only if the reward is strictly equal to 1.

**ScienceWorld**  ScienceWorld is a text-based simulation environment centered on completing basic scientific experiments. The agent needs to interact with the environment to measure certain values or to complete an experiment by applying scientific knowledge. The tasks in ScienceWorld are meticulously categorized, with each task having several variations and offering numerous different sub-goals. We adopt the data split from ETO (Song et al., 2024) and exclude Task-9 and Task-10 from the trajectories. ScienceWorld provides a dense reward between 0 and 1 for each task, based on the completion of its sub-goals.

The data statistics for each environment are summarized in Table 7.

Table 7: Data statistics for the three environments

|              | Train | Seen | Unseen |
|--------------|-------|------|--------|
| ALFWorld     | 3553  | 140  | 134    |
| WebShop      | 1824  | 200  | -      |
| ScienceWorld | 1483  | 194  | 211    |

**ManiSkill**  We evaluate embodied decision making using two manipulation tasks from ManiSkill (Tao et al., 2025), *PickCube v1* and *PullCube v1*, both implemented with the official Franka Panda robot. Each environment exposes a continuous control interface based on incremental end–effector pose and gripper commands. To make this interface compatible with LLMs, we wrap the ManiSkill environments and provide a small set of textual control actions with continuous parameters. These actions allow the LLM to issue fine grained motor commands through executable code.

We expose four textual actions that map directly to the underlying ManiSkill action vector: `move(dx, dy, dz)` specifies continuous translation of the end effector, `rotate(d_roll, d_pitch, d_yaw)` specifies continuous rotation increments, and `gripper(opening)` sets a continuous gripper opening value in $[0, 1]$. All parameters are real-valued floats and are clipped to the action bounds enforced by ManiSkill. This interface preserves the continuous nature of the control problem. Besides, to provide a compact textual observation to the LLM, we extract and serialize key environment states including the end–effector position, object position, goal position, and (for PickCube) the grasp status. This produces a consistent text description for each step while avoiding direct exposure of raw simulator states.

## A.2 PROMPTS

This appendix provides detailed prompt templates for ReAct, CodeAct and ReCode agents.

**ReAct Agent**   The ReAct agent employs a reasoning and acting cycle pattern, solving problems by alternating between thinking and acting. The following are the prompt templates used in ALFWorld, ScienceWorld, and WebShop.

---

**ALFWorld Prompt of ReAct Agent**

```
ALFWorldPrompt = """
Interact with a household to solve a task.

Your response should use the following format:
Think: I think ... or
Action:
---
Here are two examples.
{examples}
(End of examples)
---

Here is your task:
"""
```

---

**ScienceWorld Prompt of ReAct Agent**

```
SciWorldPrompt = """
You are a helpful assistant to do some scientific experiment in an environment.
In the environment, there are several rooms: kitchen, foundry, workshop, bathroom, outside, living room,
↪  bedroom, greenhouse, art studio, hallway
You should explore the environment and find the items you need to complete the experiment.
You can teleport to any room in one step.
All containers in the environment have already been opened, you can directly get items from the
↪  containers.

The available actions are:
open OBJ: open a container
close OBJ: close a container
activate OBJ: activate a device
deactivate OBJ: deactivate a device
connect OBJ to OBJ: connect electrical components
disconnect OBJ: disconnect electrical components
use OBJ [on OBJ]: use a device/item
look around: describe the current room
examine OBJ: describe an object in detail
look at OBJ: describe a container's contents
read OBJ: read a note or book
move OBJ to OBJ: move an object to a container
pick up OBJ: move an object to the inventory
pour OBJ into OBJ: pour a liquid into a container
mix OBJ: chemically mix a container
teleport to LOC: teleport to a specific room
focus on OBJ: signal intent on a task object
wait: task no action for 10 steps
wait1: task no action for a step
---
Here is the example:

{examples}
---

Now, it's your turn and here is the task.
"""
```

---

**WebShop Prompt of ReAct Agent**

```
WebShopPrompt = """
You are doing a web shopping task. I will give you instructions about what to do. You have to
follow the instructions. Every round I will give you an observation, you have to respond to an action
↪  based on the state and instruction. You can use search action if search is available. You can click
↪  one of the buttons in clickables. An action should be one of the following structure:
↪  search[keywords] or click[value].
```

```
If the action is not valid, perform nothing. Keywords in search are up to you, but the value in click
must be a value in the list of available actions. Remember that your keywords in search should be
carefully designed.

Your response should use the following format:
Think: I think ... or
Action: click[something]/search[keywords]

---

Task:
"""
```

**CodeAct Agent**   CodeAct agent solves problems by generating and executing Python code, interacting with the environment using specific functions.

---

**Prompt of CodeAct Agent**

```
CodeActPrompt = """
You are a helpful assistant assigned with the task of problem-solving. To achieve this, you will be using
↪  an interactive coding environment equipped with a variety of tool functions to assist you throughout
↪  the process.

At each turn, you have two step to finish:
1. you should first provide your step-by-step thinking for solving the task. Your thought process should
↪  be enclosed using "<thought>" tag, for example: <thought> I need to print "Hello World!" </thought>.
2. After that, you should interact with a Python programming environment and receive the corresponding
↪  output. Your code should be enclosed using "<execute>" tag, for example: <execute> print("Hello
↪  World!") </execute>.

The environment provides the following functions that you can only use:
{tool_desc}

Only use the functions above with `print()` to get environment feedback at the same time.

---
Here is an example of how to use the functions:
{in_context_example}
(End of Example)
---
"""
```

**ReCode Agent**   ReCode Agent shares a set of prompt templates across all environments, providing different primitive action descriptions and examples based on the environment.

---

**Prompt of ReCode Agent**

```
ReCodePrompt = """
You are the EXPAND step in the LLM Agent loop. You need to replace the current placeholder function node
↪  with its code implementation.

Decide how to implement the placeholder:
- If the subtask of current function can be done in 1-2 primitive actions from the list below, write them
↪  directly using `run(action: str)`.
- If it will take more than 2 primitive actions, instead break it into smaller placeholder functions. Each
↪  sub-goal should be clear, meaningful, and ordered so that completing them achieves the current task.

All legal primitive actions are:
{available_actions}
And all of them should be used in the function `run(action: str) -> str`, which returns an observation in
↪  string format.

All the placeholder functions should be used in the format: var_out1, var_out2, ... =
↪  snake_style_function_name(var_in1, var_in2="explicitly declared variables will also be registered",
↪  ...), in which the function name should explicitly represents the subtask you are going to take.
```

```
Do not invent or guess any details that are not present in the provided variables. If essential
↪   information is missing or uncertain (such as which target to use, what value to set, or which step to
↪   take next), write a descriptive placeholder function that explicitly represents the missing
↪   decision), to be expanded later.
Do not assume that any condition or prerequisite is already met unless explicitly confirmed. If something
↪   must be prepared, accessed, or changed, include explicit steps or sub-goals to do so.

In your response:
1. Start with a brief natural language explanation of how you will complete or break down the task,
↪   enculded with <think> and </think>.
2. Then output a Python code with <execute> and </execute> tags, containing only valid actions or
↪   commands for this environment. Do not create functions with `def`, and do not place placeholder
↪   functions inside loop or condition structures.

---
Here are some examples to guide the style and format, each example is ONLY ONE turn of the interaction:
{examples}
(End of Examples)
---

The current function to expand is:
{task}
The variables you can use is:
{variables}
"""
```

## A.3 FEWSHOT EXAMPLES

We provide detailed examples used by the ReCode agent across different environments. Each example demonstrates the hierarchical decomposition approach where high-level tasks are progressively expanded into executable primitive actions.

**ALFWorld Examples**  The ReCode agent uses task-specific examples for different categories in ALFWorld. Each category represents a distinct household task pattern with slight variations in logic while maintaining consistent structure. Here we present the **cool** task category as a representative example.

---

**Cool Task Examples of ALFWorld**

```
ALFWorldExamples = """
[Input]
The current function to expand is:
solve(instruction, observation)
The variables you can use is:
- instruction (str): Your task is to: put a cool mug in shelf.
- observation (str): You are in the middle of a room. Looking quickly around you, you see a cabinet 9, a
↪   cabinet 8, a cabinet 7, a cabinet 6, a cabinet 5, a cabinet 4, a cabinet 3, a cabinet 2, a cabinet 1,
↪   a coffeemachine 1, a countertop 3, a countertop 2, a countertop 1, a drawer 9, a drawer 8, a drawer
↪   7, a drawer 6, a drawer 5, a drawer 4, a drawer 3, a drawer 2, a drawer 1, a fridge 1, a garbagecan
↪   1, a microwave 1, a shelf 3, a shelf 2, a shelf 1, a sinkbasin 1, a stoveburner 4, a stoveburner 3, a
↪   stoveburner 2, a stoveburner 1, and a toaster 1.

[Output]
<think>
The task requires cooling, which must be done with the only fridge present (`fridge 1`). The final target
↪   is a shelf, and there are three shelves; I will choose `shelf 1` as an explicit target. I persist
↪   both IDs. I also create `all_location_IDs` from the initial observation. The plan decomposes into
↪   (1) find and take a mug by iterating known locations, handling closed receptacles via `open`; (2)
↪   cool the mug at the fridge after ensuring access; (3) place the mug on the chosen shelf.
</think>

<execute>
import random
cooling_location_ID = 'fridge 1'
target_location_ID = 'shelf 1'
obj = 'mug'
all_location_IDs = re.findall(r'[a-z]+(?:[a-z]+)? \d+', observation)
random.shuffle(all_location_IDs)

obj_ID = find_and_take(obj, all_location_IDs)
cool_with(obj_ID, cooling_location_ID)
```

---

```
put_in(obj_ID, target_location_ID)
</execute>

[Input]
The current function to expand is:
obj_ID = find_and_take(obj, all_location_IDs)
The variables you can use is:
- obj (str): mug
- all_location_IDs (list[str]): ['cabinet 9', 'cabinet 8', 'cabinet 7', 'cabinet 6', 'cabinet 5',
↪  'cabinet 4', 'cabinet 3', 'cabinet 2', 'cabinet 1', 'coffeemachine 1', 'countertop 3', 'countertop
↪  2', 'countertop 1', 'drawer 9', 'drawer 8', 'drawer 7', 'drawer 6', 'drawer 5', 'drawer 4', 'drawer
↪  3', 'drawer 2', 'drawer 1', 'fridge 1', 'garbagecan 1', 'microwave 1', 'shelf 3', 'shelf 2', 'shelf
↪  1', 'sinkbasin 1', 'stoveburner 4', 'stoveburner 3', 'stoveburner 2', 'stoveburner 1', 'toaster 1']

[Output]
<think>
The goal is to navigate the room methodically to find any instance of the target object and pick it up. I
↪  will iterate through the known locations, always navigating first. If a location produces no new
↪  information, I will examine it. When a location is closed, I will open it before searching. Once I
↪  observe a concrete `{obj} k` mention, I will parse that exact ID and take it from the current
↪  location. Common pitfalls include not examining after navigation (missing items), failing to open a
↪  closed container, or taking without the full `obj_ID`.
</think>

<execute>
for location_ID in all_location_IDs:
    obs = run(f'go to {location_ID}')
    if 'Nothing happens.' in obs:
        obs = run(f'examine {location_ID}')
    location_closed = 'closed' in obs
    if location_closed:
        obs = run(f'open {location_ID}')
    if f" {obj} " in obs: # Prevent names from containing relationships
        obj_ID = re.search(rf'\b{obj} \d+\b', obs).group()
        run(f'take {obj_ID} from {location_ID}')
        break
</execute>

[Input]
The current function to expand is:
cool_with(obj_ID, cooling_location_ID)
The variables you can use is:
- obj_ID (str): mug 1
- cooling_location_ID (str): fridge 1

[Output]
<think>
Cooling is only valid with a fridge. I will ensure access to the fridge: navigate and issue the cooling
↪  command with the exact object ID. Pitfalls: using the wrong station (sink/microwave) or skipping
↪  navigation.
</think>

<execute>
run(f'go to {cooling_location_ID}')
run(f'cool {obj_ID} with {cooling_location_ID}')
</execute>

[Input]
The current function to expand is:
put_in(obj_ID, target_location_ID)
The variables you can use is:
- obj_ID (str): mug 1
- target_location_ID (str): shelf 1

[Output]
<think>
```

```
I will navigate to the target location, examine if nothing happens, determine if it is closed, and open
↪  if needed. Then I will move the object in. This logic supports both containers (openable) and
↪  surfaces (non-openable, but harmless to check). Pitfalls: using the wrong ID in `examine` or
↪  skipping the `location_closed` check.
</think>

<execute>
obs = run(f'go to {target_location_ID}')
if 'Nothing happens.' in obs:
    obs = run(f'examine {target_location_ID}')
location_closed = 'closed' in obs
if location_closed:
    obs = run(f'open {target_location_ID}')
run(f'move {obj_ID} to {target_location_ID}')
</execute>
"""
```

**ScienceWorld Examples**   The ScienceWorld examples demonstrate scientific experiment tasks that require domain knowledge application and spatial reasoning. The agent must navigate between rooms, manipulate objects, and perform chemistry or physics operations. Here we present a representative chemistry task: creating green paint through color mixing.

### ScienceWorld Examples

```
SciWorldExamples = """
[Input]
The current function to expand is:
solve(instruction, observation)
The variables you can use is:
- instruction (str): Your task is to use chemistry to create green paint. When you are done, focus on the
↪  green paint.
- observation (str): This room is called the bedroom. In it, you see:
the agent
a substance called air
a bed. On the bed is: a mattress. On the mattress is: a white pillow.
a closet. The closet door is open. In the closet is: nothing.
a picture
a table. On the table is: nothing.
You also see:
A door to the hallway (that is open)

[Output]
<think>
I need to find materials to create green paint, which are likely to be found in an art studio. The
↪  bedroom doesn't have the materials I need, so I should go to a place where paint and painting
↪  supplies are likely to be found.
</think>

<execute>
run("teleport to art studio")
obs = run("look around")
solve(instruction, obs)
</execute>

[Input]
The current function to expand is:
solve(instruction, obs)
The variables you can use is:
- instruction (str): Your task is to use chemistry to create green paint. When you are done, focus on the
↪  green paint.
- obs (str): This room is called the art studio. In it, you see:
the agent
a substance called air
a large cupboard. The large cupboard door is open. In the large cupboard is: a drawer, a wood cup
↪  (containing yellow paint), a wood cup (containing blue paint), a wood cup (containing red paint).
a table. On the table is: a glass cup (containing nothing).
a wood cup (containing yellow paint)
a wood cup (containing blue paint)
a wood cup (containing red paint)
You also see:
A door to the hallway (that is open)
```

```
[Output]
<think>
Now that I'm in the art studio, I can see blue paint and yellow paint which are the primary colors needed
↪ to mix green paint. I should call the mix_blue_and_yellow_paints function to handle the mixing
↪ process, then look around to see the result, and finally focus on the green paint.
</think>

<execute>
mix_blue_and_yellow_paints(obs)
obs = run("look around")
focus_on_green_paint(obs)
</execute>

[Input]
The current function to expand is:
mix_blue_and_yellow_paints(obs)
The variables you can use is:
- obs (str): This room is called the art studio. In it, you see:
the agent
a substance called air
a large cupboard. The large cupboard door is open. In the large cupboard is: a drawer, a wood cup
↪ (containing yellow paint), a wood cup (containing blue paint), a wood cup (containing red paint).
a table. On the table is: a glass cup (containing nothing).
a wood cup (containing yellow paint)
a wood cup (containing blue paint)
a wood cup (containing red paint)
You also see:
A door to the hallway (that is open)

[Output]
<think>
I can see blue paint and yellow paint available. I need to follow the chemistry process: first pour the
↪ blue paint into the empty glass cup, then add yellow paint to the same container, and finally mix
↪ them together to create green paint. This follows the principle that blue and yellow are primary
↪ colors that combine to make green.
</think>

<execute>
run("pour cup containing blue paint in art studio into cup containing nothing")
obs = run("pour cup containing yellow paint in art studio in cup containing blue paint in table")
obs = run("mix cup containing blue paint and yellow paint")
</execute>

[Input]
The current function to expand is:
focus_on_green_paint(obs)
The variables you can use is:
- obs (str): This room is called the art studio. In it, you see:
the agent
a substance called air
a large cupboard. The large cupboard door is open. In the large cupboard is: a drawer, a wood cup
↪ (containing yellow paint), a wood cup (containing blue paint), a wood cup (containing red paint).
a table. On the table is: a glass cup (containing green paint).
a wood cup (containing yellow paint)
a wood cup (containing blue paint)
a wood cup (containing red paint)
You also see:
A door to the hallway (that is open)

[Output]
<think>
Perfect! I can see that the glass cup now contains green paint. The task is to create green paint and
↪ then focus on it. Now that the green paint is made, I should concentrate on the final product.
</think>

<execute>
run("focus on green paint")
</execute>
"""
```

**WebShop Examples**   The WebShop examples demonstrate e-commerce navigation tasks where the agent must search for products, evaluate options against specified requirements, and complete purchases. Note that since clickable buttons in the environment are represented with square brackets (e.g., [Search], [Buy Now]), we use INPUT: and OUTPUT: instead of [Input] and [Output] in these examples to avoid confusion.

---

**WebShop Examples**

```
WebShopExamples = """
INPUT:
The current function to expand is:
solve(instruction, observation)
The variables you can use is:
- instruction (str): i need a long lasting 6.76 fl oz bottle of l'eau d'issey, and price lower than
↪  100.00 dollars
- observation (str): i need a long lasting 6.76 fl oz bottle of l'eau d'issey, and price lower than
↪  100.00 dollars
[Search]

OUTPUT:
<think>
I need a 6.76 fl oz bottle of L'eau d'Issey perfume with a price lower than $100. I need to:
1. First search for relevant products
2. Click to view product details
3. Decide whether to buy or exclude based on price and specification requirements

Let me execute this task step by step.
</think>

<execute>
search_obs = search_with_key_words(instruction, observation)
item_id, detail_obs = click_item(search_obs, instruction)
exclude_or_buy(search_obs, detail_obs, instruction)
</execute>

INPUT:
The current function to expand is:
search_obs = search_with_key_words(instruction, observation)
The variables you can use is:
- instruction (str): i need a long lasting 6.76 fl oz bottle of l'eau d'issey, and price lower than
↪  100.00 dollars
- observation (str): i need a long lasting 6.76 fl oz bottle of l'eau d'issey, and price lower than
↪  100.00 dollars
[Search]

OUTPUT:
<think>
I need to use the search function to find products that meet the criteria: 6.76 fl oz bottle of L'eau
↪  d'Issey with price lower than $100. The search keywords should include the product name, size
↪  specification, and price constraint.
</think>

<execute>
search_obs = run("search[l'eau d'issey 6.76 fl oz bottle price < 100.00]")
</execute>

INPUT:
The current function to expand is:
item_id, detail_obs = click_item(search_obs, instruction)
The variables you can use is:
- search_obs (str): [Back to Search]
Page 1 (Total results: 50)
[Next >]
[B000VOHH8I]
L'eau D'issey By Issey Miyake for MenEau De Toilette Spray, 6.7 Fl Oz Bottle
$64.98
[B000MJZOPK]
L'eau d'Issey by Issey Miyake for Women 3.3 oz Eau de Toilette Spray
$49.98
[B0012S249E]
```

```
L'eau D'issey By Issey Miyake For Women. Shower Cream 6.7-Ounces
$31.36
[B01H8PGKZS]
L'eau D'Issey FOR MEN by Issey Miyake - 6.7 oz EDT Spray
$67.97
[B00G3C8FHE]
L'Eau d'Issey pour Homme - Eau de Toilette 4.2 fl oz
$51.25
[B000R94HRG]
Issey Miyake L'Eau D'Issey Pour Homme Eau De Toilette Natural Spray
$44.99
[B000C214CO]
Issey Miyake L'eau D'issey Eau de Toilette Spray for Men, 4.2 Fl Oz
$53.99
[B0018SBRDC]
Issey Miyake L'eau d'Issey for Women EDT, White, 0.84 Fl Oz
$27.04
[B000XEAZ9Y]
L'eau De Issey By Issey Miyake For Men. Eau De Toilette Spray 6.7 Fl Oz
$67.08
[B079HZR2RX]
L'eau d'Issey Pure by Issey Miyake for Women 3.0 oz Nectar de Parfum Spray
$71.49
- instruction (str): i need a long lasting 6.76 fl oz bottle of l'eau d'issey, and price lower than
↪  100.00 dollars

OUTPUT:
<think>
From the search results, I can see multiple L'eau d'Issey products. I need to choose the one that best
↪  matches my requirements:
- I need 6.76 fl oz size
- Price should be lower than $100

Looking at the search results, product B000VOHH8I is "L'eau D'issey By Issey Miyake for Men Eau De
↪  Toilette Spray, 6.7 Fl Oz Bottle" priced at $64.98. This is the closest to my required 6.76 fl oz
↪  size and is under $100. I should click on this product to view its details.
</think>

<execute>
item_id = "B000VOHH8I"
detail_obs = run(f"click[{item_id}]")
</execute>

INPUT:
The current function to expand is:
exclude_or_buy(search_obs, detail_obs, instruction)
The variables you can use is:
- search_obs (str): [Back to Search]
Page 1 (Total results: 50)
[Next >]
[B000VOHH8I]
L'eau D'issey By Issey Miyake for MenEau De Toilette Spray, 6.7 Fl Oz Bottle
$64.98
...
- detail_obs (str): [Back to Search]
[< Prev]
size
[2.5 fl oz]
[6.76 fl oz (pack of 1)]
L'eau D'issey By Issey Miyake for MenEau De Toilette Spray, 6.7 Fl Oz Bottle
Price: $64.98
Rating: N.A.
[Description]
[Features]
[Reviews]
[Buy Now]
- instruction (str): i need a long lasting 6.76 fl oz bottle of l'eau d'issey, and price lower than
↪  100.00 dollars

OUTPUT:
<think>
I'm looking at the product details, which shows L'eau D'issey By Issey Miyake for Men Eau De Toilette
↪  Spray. My requirements are:
1. 6.76 fl oz bottle size
2. L'eau d'Issey brand
```

```
3. Price lower than $100

This product meets all requirements:
- The exact 6.76 fl oz size option is available
- It's the correct brand
- The price $64.98 is well under $100

I should select the 6.76 fl oz size option and proceed with the purchase.
</think>

<execute>
run("click[6.76 fl oz (pack of 1)]")
run("click[Buy Now]")
</execute>
"""
```

## A.4 QUANTITATIVE ANALYSIS OF ReCode

To provide quantitative evidence for how ReCode adjusts its decision granularity during execution, we report statistics computed over trajectories on the unseen splits from ALFWorld, ScienceWorld and WebShop using GPT-4o mini. These metrics characterize the hierarchical structure that emerges from ReCode's recursive expansion process.

We measure three structural properties of the generated decision trees and these metrics capture the balance between coarse and fine decisions across tasks.

- Average recursion depth, defined as the maximum expansion depth reached in an episode, averaged over all trajectories;
- Average branching factor, defined as the average number of child functions or substeps produced by each expansion;
- Placeholder ratio, defined as the fraction of generated functions that remain as high-level placeholders rather than being expanded into primitive actions.

| Environment | Avg. Depth ($\mu \pm \sigma$) | Avg. Branching ($\mu \pm \sigma$) | Placeholder Ratio |
|---|---|---|---|
| ALFWorld | $1.93 \pm 0.45$ | $3.87 \pm 2.36$ | 0.19 |
| WebShop | $3.97 \pm 0.65$ | $5.68 \pm 3.71$ | 0.17 |
| ScienceWorld | $5.60 \pm 3.34$ | $8.02 \pm 5.50$ | 0.11 |

Table 8: Quantitative statistics of ReCode's decision structures on the unseen splits from ALFWorld, ScienceWorld and WebShop using GPT-4o mini.

**Discussion** The statistics reveal clear differences in how ReCode structures its decisions across environments. ALFWorld has the smallest average depth and branching factor, suggesting that its tasks rarely require deeply nested plans and instead emphasize structured exploration and local interaction in relatively simple scenes. WebShop uses deeper trees with larger branching factors, which is consistent with multi step product search, filtering, and comparison that naturally induce several layers of decision points. ScienceWorld exhibits the deepest and most complex decision structures, reflecting its multi stage experimental procedures and frequent decomposition into intermediate subgoals. Across all environments, the placeholder ratios remain relatively low and tend to decrease as task complexity increases. A plausible explanation is that more complex tasks require richer environment dynamics to support reliable planning, so ReCode executes more primitive actions and interacts more frequently with the environment before or alongside high level abstractions. These patterns provide quantitative evidence that ReCode adapts its decision structure and effective granularity to the structure and complexity of each environment.

## A.5 CASE STUDY

Initially, the agent formulates a hybrid plan that integrates abstract sub-tasks with concrete, primitive actions. This initial plan, shown in the following code block, includes distinct sub-procedures like

find_and_take for complex procedures that require further decomposition, while directly embedding known, simple steps like run('go to dresser 1') as primitive actions within the main strategic flow.

The agent then executes this hybrid plan sequentially. Primitive actions like run(...) are executed directly by the environment interpreter. In contrast, placeholder functions such as find_and_take trigger an on-demand expansion, where the agent generates the necessary code to complete that specific sub-task just in time. This hybrid strategy allows the agent to maintain a high-level, strategic view of the task while committing to concrete actions only when the steps are simple and certain. It combines the robustness of hierarchical planning with the efficiency of direct execution, providing a flexible and powerful problem-solving framework that can adapt its level of abstraction as needed.

```python
# ... initial setup and variable declaration
    ...
all_location_IDs = declare_init_vars(
    instruction, observation)

# Decompose task: find and place the first
    alarm clock
obj_ID, all_location_IDs = find_and_take('
    alarmclock', all_location_IDs)
run('go to dresser 1')
put_in_again(obj_ID, 'dresser 1')

# Decompose task: find and place the second
    alarm clock
obj_ID, all_location_IDs = find_and_take('
    alarmclock', all_location_IDs)
run('go to dresser 1')
put_in_again(obj_ID, 'dresser 1')
```

To provide a concrete illustration of ReCode's decision-making process, we analyze a representative trajectory from the ALFWorld environment for the task "put two alarmclock in dresser". The agent's complete execution flow, derived from the log file, is detailed below. Figure 5 provides a high-level visualization of this recursive process.

The agent's primary goal is to place two alarm clocks in a dresser. Given that it can only carry one object at a time, a sequential find-and-place strategy is necessary.

The following steps detail the agent's turn-by-turn execution. It is important to note that this is not a pre-generated script. Instead, the agent starts with the high-level plan from Step 1 and executes it line by line. When the executor encounters an unimplemented placeholder function (e.g., declare_init_vars), execution pauses, and a recursive call is made to the LLM to generate the necessary code for that specific function. The newly generated code is then executed, and this cycle repeats until the entire plan is resolved.

**Step 1: Initial High-Level Plan** First, the agent formulates an initial high-level plan by expanding the root solve function. This plan breaks down the complex task into a sequence of abstract placeholder functions without specifying low-level primitive actions. This strategic scaffold includes steps for finding and placing each of the two alarm clocks, as well as updating its internal knowledge about object locations.

```python
# Initial plan generated by expanding solve(...)
obj, target_location_ID, all_location_IDs = declare_init_vars(instruction
    , observation)

# Decompose task: find and place the first alarm clock
obj_ID, location_ID = find_and_take(obj, all_location_IDs)
put_in(obj_ID, target_location_ID)

# Decompose task: find and place the second alarm clock
all_location_IDs = update_all_location_IDs(location_ID,
    target_location_ID, all_location_IDs)
obj_ID = find_and_take_again(obj, all_location_IDs)
put_in_again(obj_ID, target_location_ID)
```

**Step 2: Declaring Initial Variables** The agent begins by executing the first placeholder, declare_init_vars. It expands this function to parse the initial instruction and observation, identifying key variables. This step grounds the agent by establishing the target object ("alarmclock"), the destination ("dresser 1"), and a comprehensive list of all searchable locations.

```python
# Code generated by expanding declare_init_vars(...)
target_location_ID = 'dresser 1'
```

```
In[1]: solve(instruction, observation)
- instruction (str): put two alarmclock
in dresser
- observation (str): You are in the
middle of a room. Looking quickly
around you, you see a bed 1, a desk 1,
a drawer 5, a drawer 4, a drawer 3, a
drawer 2, a drawer 1, a dresser 1, a
garbagecan 1, a laundryhamper 1, and a
shelf 1.

Out[1]:
obj, target_location_ID,
all_location_IDs =
declare_init_vars(instruction,
observation)

obj_ID, location_ID
= find_and_take(obj, all_location_IDs)
put_in(obj_ID, target_location_ID)

all_location_IDs =
update_all_location_IDs(location_ID,
target_location_ID, all_location_IDs)

obj_ID = find_and_take_again(obj,
all_location_IDs)
put_in_again(obj_ID,
target_location_ID)
```

```
In[2]: ...
Out[2]:
target_location_ID = 'dresser 1'
obj = 'alarmclock'
all_location_IDs = ['bed 1', ...]
```

```
...
```

```
In[5]: all_location_IDs =
update_all_location_IDs(location_ID,
target_location_ID, all_location_IDs)
- location_ID (str): desk 1
- target_location_ID (str): dresser 1
- all_location_IDs (list[str]): [...]
Out[6]:
idx = all_location_IDs.index(location_ID)
all_location_IDs = all_location_IDs[idx:]
if target_location_ID in all_location_IDs:
    all_location_IDs.remove(...)
```

```
In[6]: obj_ID =
find_and_take_again(obj,
all_location_IDs)
- obj (str): alarmclock
- all_location_IDs (list[str]):
['desk 1', ...]
Out[6]:
for location_ID in all_location_IDs:
    obs = run(f'go to {location_ID}')
    if 'Nothing happens.' in obs:
        obs = run(f'examine
{location_ID}')
    location_closed = 'closed' in obs
    if location_closed:
        obs = run(f'open
{location_ID}')
    if f" {obj} " in obs:
        obj_ID = re.search(rf'\b{obj}
\d+\b', obs).group()
        run(f'take {obj_ID} from
{location_ID}')
        break
```

```
In[7]: put_in_again(obj_ID, target_location_ID)
- obj_ID (str): alarmclock 2
- target_location_ID (str): dresser 1
Out[7]:
obs = run(f'go to {target_location_ID}')
if 'Nothing happens.' in obs:
    obs = run(f'examine {target_location_ID}')
location_closed = 'closed' in obs
if location_closed:
    obs = run(f'open {target_location_ID}')
run(f'move {obj_ID} to {target_location_ID}')
```

Figure 5: A visualization of the ReCode execution flow for the task "put two alarmclock in dresser" in ALFWorld. The diagram shows how a high-level plan, composed of placeholder functions (e.g., find_and_take_again), is recursively expanded on demand. Each arrow points from a function call to the code block it generates. This process illustrates the dynamic transition from abstract planning to the generation of fine-grained, executable code.

```
obj = 'alarmclock'
all_location_IDs = ['bed 1', 'desk 1', 'drawer 5', 'drawer 4', 'drawer 3'
    , 'drawer 2', 'drawer 1', 'dresser 1', 'garbagecan 1', 'laundryhamper
    1', 'shelf 1']
```

**Step 3: Finding and Taking the First Alarm Clock**  Upon successfully executing the variable declarations, the executor proceeds to the next line in the plan: the find_and_take placeholder. This triggers another on-demand expansion cycle, prompting the agent to generate the following code for a methodical search:

```
# Code generated by expanding find_and_take(...)
for location_ID in all_location_IDs:
    obs = run(f'go to {location_ID}')
    if 'Nothing happens.' in obs:
        obs = run(f'examine {location_ID}')
    location_closed = 'closed' in obs
    if location_closed:
```

```
        obs = run(f'open {location_ID}')
    if f" {obj} " in obs:
        obj_ID = re.search(rf'\b{obj} \d+\b', obs).group()
        run(f'take {obj_ID} from {location_ID}')
        break

# Key environment feedback during execution:
# You arrive at desk 1. On the desk 1, you see a alarmclock 3, a
    alarmclock 2...
# You pick up the alarmclock 3 from the desk 1.
```

**Step 4: Placing the First Alarm Clock** After the code from the previous step successfully executes and finds "alarmclock 3", the agent continues to the next task in its high-level plan, put_in. This function is then expanded to generate the code for navigating to the dresser and placing the object inside.

```
# Code generated by expanding put_in(...)
obs = run(f'go to {target_location_ID}')
if 'Nothing happens.' in obs:
    obs = run(f'examine {target_location_ID}')
location_closed = 'closed' in obs
if location_closed:
    obs = run(f'open {target_location_ID}')
run(f'move {obj_ID} to {target_location_ID}')

# Environment feedback:
# You move the alarmclock 3 to the dresser 1.
```

**Step 5: Updating Location Knowledge** After placing the first clock, the agent must update its search space for the second clock. It expands update_all_location_IDs, which intelligently slices the location list to resume searching from the last successful spot ("desk 1") and removes the destination ("dresser 1") to prevent redundant checks.

```
# Code generated by expanding update_all_location_IDs(...)
# location_ID is 'desk 1' from the previous step.
all_location_IDs = all_location_IDs[all_location_IDs.index(location_ID):]
if target_location_ID in all_location_IDs:
    all_location_IDs.remove(target_location_ID)

# Resulting all_location_IDs for next search:
# ['desk 1', 'drawer 5', ..., 'shelf 1']
```

**Step 6 & 7: Finding and Placing the Second Alarm Clock** Finally, the agent repeats the find-and-place cycle by expanding find_and_take_again and put_in_again. It reuses the same logic as before but operates on the updated location list. It successfully finds "alarmclock 2" on "desk 1", picks it up, navigates back to the dresser, and places it inside, completing the task.

```
# Agent expands find_and_take_again(...) and finds the second clock.
# Environment feedback:
# You arrive at desk 1. On the desk 1, you see a alarmclock 2, a
    alarmclock 1...
# You pick up the alarmclock 2 from the desk 1.

# Agent expands put_in_again(...) and places the second clock.
# Environment feedback:
# You move the alarmclock 2 to the dresser 1.
```

