# OpenReview forum: "ReCode: Unify Plan and Action for Universal Granularity Control"
_ICLR.cc/2026/Conference — Submitted to ICLR 2026_

### Official Review · Reviewer_VZQm · 2025-10-26

**Soundness:** 3
**Presentation:** 2
**Contribution:** 3
**Rating:** 6
**Confidence:** 3

**Summary:**

This paper proposes ReCode, a recursive code generation paradigm for LLM-based agents that unifies planning and action within a single executable code framework. The key idea is to treat planning as high-level action, **recursively** decomposing abstract placeholder functions into executable API calls until primitive actions are reached. This approach aims to enable dynamic granularity control, eliminate the need for predefined action spaces, and generate rich hierarchical training data. The authors evaluate ReCode across three text-based environments (ALFWorld, WebShop, ScienceWorld) and demonstrate improvements in inference performance, generalization, and training efficiency compared to strong baselines like ReAct and CodeAct.

**Strengths:**

1. This approach of initially employing placeholders for complex actions and iteratively refining them into atomic actions or more detailed plans is insightful.
2. Both plans and actions are represented as Python function calls (placeholders, atomic), which are generated and executed by a single policy $\pi$, thereby eliminating the conventional separation between planning and execution. This is conceptually elegant and practically implementable.
3. Figure 2 offers an intuitive illustration. Once you see it, you pretty much get how ReCode works.
4. The Cost Analysis effectively demonstrates the cost-efficient nature of the ReCode framework.

**Weaknesses:**

1. About the writing, it would be better to just clearly list out the specific contributions in the Introduction.
2. Algorithm 1 mentions using a HeuristicConvert function to turn task instruction into a root. I feel like this part isn't really explained—it's not clear how it actually works.
3. All three are text-world benchmarks. This seems a bit limited. I'm not totally convinced it would work in more complex settings. I'd like to see how it performs on more diverse and harder benchmarks, like EXP-Bench [1], or RExBench [2].

[1] Kon, P. T. J., Liu, J., Zhu, X., Ding, Q., Peng, J., Xing, J., ... & Chen, A. (2025). EXP-Bench: Can AI Conduct AI Research Experiments?. arXiv preprint arXiv:2505.24785.

[2] Edwards, N., Lee, Y., Mao, Y. A., Qin, Y., Schuster, S., & Kim, N. (2025). RExBench: Can coding agents autonomously implement AI research extensions?. arXiv preprint arXiv:2506.22598.

**Questions:**

1. When a placeholder expansion fails (invalid code, violated state assumptions), is there any recovering or replanning mechanism? Can you quantify the recovery rate and its extra cost?
2. About the HeuristicConvert part, could you share some insights into how it was designed? It would be great to see an ablation study testing if different heuristics would change the results.
3. Can you conduct experiments on EXP-bench, or RExBench?
4. For Table 2, could you also add the results for AdaPlanner and ADaPT using models like Gemini 2.5 Flash and DeepSeek-V3.1 on these three environments?

---

> ### Author Response · Authors · 2025-11-21
> **Response to Reviewer VZQm**
>
> We thank the reviewer VZQm for raising constructive suggestions. Please find our response below.
>
> > W1. About the writing, it would be better to just clearly list out the specific contributions in the Introduction.
>
> Thank you for the suggestion. In the revised version, we added a concise bullet list of contributions in the end of **Section 1 Introduction** to make the paper’s key contributions clearer.
>
> > W2. & Q2. Algorithm 1 mentions using a HeuristicConvert function to turn task instruction into a root. I feel like this part isn't really explained—it's not clear how it actually works.
>
> Thank you for pointing this out. In the revised version, we updated the **Figure 2** and expanded the corresponding description in **Section 3.3** to clearly explain how the task instruction is converted into the initial root function. We also renamed the component to avoid the misleading term “HeuristicConvert”, since this step is simply a direct text-to-code transformation rather than a heuristic procedure.
>
> > Q1. When a placeholder expansion fails (invalid code, violated state assumptions), is there any recovering or replanning mechanism? Can you quantify the recovery rate and its extra cost?
>
> Thank you for the question. As described in the revised **Section 3.3**, if a placeholder expansion produces invalid code or violates state assumptions, ReCode simply **re-expands** that local placeholder with the error message while keeping the rest of the plan unchanged, and a recursion depth cap ensures safe termination if retries fail. In practice, such failures almost rarely occur because our prompts and few-shot examples strongly constrain the code format and state assumptions, so this recovery mechanism is rarely triggered and incurs negligible extra cost.
>
> > Q4. For Table 2, could you also add the results for AdaPlanner and ADaPT using models like Gemini 2.5 Flash and DeepSeek-V3.1 on these three environments?
>
> We have added the requested multi-model results for AdaPlanner and ADaPT using Gemini 2.5 Flash and DeepSeek-V3.1 on all three environments in **Table 2**. ReCode still achieves the highest average performance across backbone models.
>
> > W3. & Q3. I'd like to see how it performs on more diverse and harder benchmarks, like EXP-Bench, or RExBench.
>
> Thank you for the suggestion. We agree that harder, more diverse benchmarks are valuable, but our focus in this paper is on the **decision-making paradigm** rather than on **a fully engineered agent stack**.
> - Benchmarks like EXP-Bench and RExBench are typically used to evaluate mature agent systems (e.g., OpenHands) with **carefully engineered tool suites, long-horizon context handling, and complex orchestration**, which go well beyond the scope of our current framework.
> - In this work we therefore compare ReCode to other paradigms such as ReAct, under a controlled, shared interface, which we believe is a fairer setting for isolating the effect of the decision mechanism itself.
>
> ReCode already demonstrates strong performance at the paradigm level, and we believe it has clear potential to serve as the core of more complete agent systems once the surrounding engineering is added, similar to how ReAct has been extended into full-featured systems like OpenHands. However, developing such a full agent stack is beyond the scope of this paper.

---

> > ### Comment · Reviewer_VZQm · 2025-11-25
> >
> > Thank you for your detailed response. I think my questions are solved. I will keep my score as 6 (marginally above the acceptance threshold).

---

### Official Review · Reviewer_wfpU · 2025-10-29

**Soundness:** 3
**Presentation:** 3
**Contribution:** 2
**Rating:** 4
**Confidence:** 4

**Summary:**

This paper introduces ReCode, a novel paradigm for LLM-based agents that unifies planning and action through recursive code generation. The key insight is that planning and action are not fundamentally different but represent decisions at different levels of abstraction. ReCode represents both plans and actions as executable code, with high-level plans as placeholder functions that are recursively refined into primitive executable actions. The authors demonstrate that this approach achieves superior performance across three benchmark environments (ALFWorld, ScienceWorld, WebShop) with over 20.9% improvement in inference and remarkable training efficiency.

**Strengths:**

- The paper is well-motivated and easy to read
- ReCode achieves remarkable cost efficiency and training efficience. This is a significant practical advantage.

**Weaknesses:**

- While the three environments are diverse, they are all text-based simulation environments. The approach needs validation on more complex, real-world tasks or environments with continuous action spaces. According to the paper's content, it seems possible to have primitive actions at much lower levels of API (for example, continuous actions like moving forward 3.5m). Experiments on this aspect are needed.
- The paper doesn't discuss how the system handles errors in code generation or execution failures. If errors occur in some parts of the code, it's questionable whether the task can be performed normally. If not, can this methodology effectively respond when LLM performance decreases?

**Questions:**

- In Section 4.2, we can see that AdaPlanner has the second-best performance. Can we see the results of testing AdaPlanner based on Gemini-Flash or DeepSeek?
- What is the maximum recursion depth observed in practice, and how does performance degrade with increasing task complexity?
- Can you provide analysis on failure modes? When does the recursive expansion fail to converge to executable actions?

---

> ### Author Response · Authors · 2025-11-21
> **Response to Reviewer wfpU 1/2**
>
> We sincerely thank the reviewer wfpU for the careful reading and valuable insights.
>
> > W1. While the three environments are diverse, they are all text-based simulation environments. The approach needs validation on more complex, real-world tasks or environments with continuous action spaces. According to the paper's content, it seems possible to have primitive actions at much lower levels of API (for example, continuous actions like moving forward 3.5m). Experiments on this aspect are needed.
>
> We appreciate the concern about validation beyond text based simulations and agree that environments with continuous action spaces are important.
> - ReCode is modality agnostic and can directly operate on low level APIs with real valued parameters, so primitive actions such as small continuous motions are naturally expressible in our framework.
> - In the revised version, we add experiments on the ManiSkill manipulation benchmark in **Section 4.4**, where control is fully continuous through delta pose and gripper commands exposed as textual APIs.
> - As shown in the new results, ReCode clearly **outperforms** ReAct and Code-as-Policies, providing concrete evidence that its recursive mechanism and granularity control extend to settings that require fine grained continuous actions.
>
> | Model            | Method | PickCube | PullCube |
> |------------------|--------|----------|----------|
> | **GPT-4o mini**  | ReAct  | 0        | 2        |
> |                  | CaP    | 34       | 34       |
> |                  | ReCode | 78       | 46       |
> | **Gemini 2.5 Flash** | ReAct  | 40       | 30       |
> |                  | CaP    | 82       | 44       |
> |                  | ReCode | 90       | 100      |
>
> > Q1. Can we see the results of testing AdaPlanner based on Gemini-Flash or DeepSeek?
>
> We have added the requested AdaPlanner results for both Gemini-Flash and DeepSeek in **Section 4.2**. These additional experiments show the same trend that ReCode consistently maintains higher average performance across all backbone models. This confirms that our conclusions are not tied to a specific model choice and that ReCode’s improvements stem from the paradigm itself.

---

> > ### Author Response · Authors · 2025-11-21
> > **Response to Reviewer wfpU 2/2**
> >
> > > W2. & Q3.
> > > The paper doesn't discuss how the system handles errors in code generation or execution failures. If errors occur in some parts of the code, it's questionable whether the task can be performed normally. If not, can this methodology effectively respond when LLM performance decreases?
> > > Can you provide analysis on failure modes? When does the recursive expansion fail to converge to executable actions?
> >
> > The revised **Section 3.3** now explicitly describes how ReCode handles code-generation errors and execution failures. ReCode expands and executes code incrementally instead of emitting a long static program, so an error only affects the current placeholder, which we simply **regenerate**.
> >
> > **Section 3.3** also clarifies the main failure mode. Recursive expansion fails to converge only when the model repeatedly produces invalid or inconsistent refinements for the same placeholder. This case is controlled by a recursion-depth cap and by strict format validation, which together guarantee safe termination. In practice, such non-convergent cases are rare.
> >
> > > Q2. What is the maximum recursion depth observed in practice, and how does performance degrade with increasing task complexity?
> >
> > In our experiments, "task complexity" refers to environments requiring more intermediate subgoals and longer reasoning chains, with ScienceWorld being more complex than ALFWorld and WebShop.
> > - As shown in **Appendix A.4**, recursion depth is globally capped but remains well below this limit in practice, typically around 2–3 in ALFWorld, 4–5 in WebShop, and 7–8 in ScienceWorld
> > - The depth ablation in **Figure 3** shows an inverted-U pattern. Moderate increases help on harder tasks, while overly deep recursion introduces unnecessary decomposition and yields a mild performance drop. This indicates that ReCode benefits from controlled recursion rather than arbitrarily deep structures.

---

> ### Comment · Reviewer_wfpU · 2025-11-25
>
> Thank you for your detailed response. My concerns have been addressed. Therefore, I will raise my score.

---

### Official Review · Reviewer_23Bv · 2025-10-31

**Soundness:** 3
**Presentation:** 3
**Contribution:** 2
**Rating:** 4
**Confidence:** 4

**Summary:**

This work proposes ReCode, a single-policy, recursive “code as plan and action” framework that refines unimplemented placeholders into executable steps, enabling flexible control over reasoning granularity. Across multiple simulated environments, it outperforms code-reasoning baselines and its hierarchical trajectories further strengthen supervised fine-tuning; however, the novelty over prior recursive code generation works (e.g., REPL-Plan) remains unclear without head-to-head comparisons.

**Strengths:**

*  **Simple, unified mechanism.** Both plans  and actions are written as code, and the system only recurses when something isn’t executable, easy to reason about and implement.
* **Gains at lower cost.** Across benchmarks, the method outperforms baselines at lower cost.

**Weaknesses:**

* The novelty over prior similar recursive code work is unclear; it could be better to add a brief comparison to previous works(e.g., REPL-Plan, Code-as-Policies) to make the contribution explicit.
*  No granulity measurement and analysis.

**Questions:**

* Could you share distributions of recursion depth and the number of expanded placeholders per episode?

* Could you add a brief, explicit paragraph contrasting ReCode with other recursive code generation papers, i.e., REPL-Plan and Code-as-Policies? A small comparison table would make the distinctions especially clear.

---

> ### Author Response · Authors · 2025-11-21
> **Response to Reviewer 23Bv**
>
> We appreciate the reviewer 23Bv's time and helpful suggestions.
>
> > W1. & Q2.
> > The novelty over prior similar recursive code work is unclear; it could be better to add a brief comparison to previous works(e.g., REPL-Plan, Code-as-Policies) to make the contribution explicit.
> > Could you add a brief, explicit paragraph contrasting ReCode with other recursive code generation papers, i.e., REPL-Plan and Code-as-Policies? A small comparison table would make the distinctions especially clear.
>
> We have added an explicit comparison with Code-as-Policies and REPL-Plan in **Section 2**, and summarize the distinctions here.
> - **Code-as-Policies (CaP)** builds a program sketch recursively **before execution**, and the resulting structure becomes a fixed policy.
> - **REPL-Plan** uses recursion mainly for subproblem isolation, but each subtask is still solved through step-by-step ReAct-style reasoning.
> - **ReCode** uses recursion as the core decision process, where each expansion generates a complete plan for the current node based **on the latest context** rather than only producing the next step.
>
> This top-down full-subtask expansion enables unified and adaptive granularity control during execution, which neither CaP’s static programs nor REPL-Plan’s step-by-step loops can provide.
>
> > W2. & Q1.
> > No granularity measurement and analysis.
> > Could you share distributions of recursion depth and the number of expanded placeholders per episode?
>
> We appreciate the suggestion to quantify granularity. In the revised version:
> - **Section 4.2** includes an ablation over maximum recursion depth, showing that recursion depth directly influences average reward and providing an operational view of granularity.
> - **Appendix A.4** further reports structural statistics of the generated decision trees across all three environments, including average depth, branching factor, and placeholder ratio.
>
> These analyses together offer a clear quantitative perspective on how ReCode adjusts its decision granularity across tasks.

---

### Official Review · Reviewer_SKxJ · 2025-11-01

**Soundness:** 2
**Presentation:** 3
**Contribution:** 2
**Rating:** 4
**Confidence:** 3

**Summary:**

This paper presents ReCode, a framework that represents an policy as a recursive code generator, unifying planning and action within a single paradigm. High-level tasks are expressed as abstract placeholder functions that the LLM recursively expands into finer-grained subfunctions and primitive actions, enabling dynamic control over decision granularity.

**Strengths:**

ReCode unifies high-level planning and low-level action within a single code-based framework, allowing LLM agents to dynamically adjust decision granularity.

**Weaknesses:**

1. The proposed approach is closely related to recent advances in code-as-policies paradigms that leverage code-generation LLMs (e.g., [1–5]). However, the paper does not sufficiently analyze or position ReCode against these methods in the related work section.

[1] Code as Policies: Language Model Programs for Embodied Control. ICRA 2023.

[2] RoboCodex: Multimodal Code Generation for Robotic Behavior Synthesis. ICML 2024.

[3] PoAct: Policy and Action Dual-Control Agent for Generalized Applications. arXiv  2025.

[4] Executable Code Actions Elicit Better LLM Agents. ICML 2024

[5] Towards Reliable Code-as-Policies: A Neuro-Symbolic Framework for Embodied Task Planning, arxiv 2025

2. The recursive expansion and execution of functions or subtasks in ReCode resemble hierarchical code-as-policies frameworks previously explored in works [6, 7]. Since these studies also pursue recursive, code-centric planning structures, the paper needs to articulate more clearly what distinguishes ReCode’s recursive granularity control from prior hierarchical code-generation methods. Moreover, given the conceptual similarity, it would be valuable to include empirical comparisons with particularly [7], one unifying reasoning and acting through executable code with adaptive feedback during execution. Technical novelties should be clearly specified.

[6] Demo2Code: From Summarizing Demonstrations to Synthesizing Code via Extended Chain-of-Thought. NeurIPS 2023.

[7] Interactive and Expressive Code-Augmented Planning with Large Language Models.  arXiv 2024.

3. While the paper claims that flexible control of decision granularity leads to superior adaptability and efficiency, the current experiment results do not isolate granularity as a directly measurable factor. To substantiate this claim, ablation experiments varying the recursive depth could demonstrate how granularity impacts reward, cost, and performance–efficiency trade-offs.

4. Beyond qualitative case studies, the paper would present quantitative analyses that reveal how granularity dynamically changes during execution. Statistics such as the average depth of generated decision trees, branching factors, and the ratio of placeholder functions to primitive actions across different tasks would provide stronger empirical support for the proposed mechanism.

5. While ReCode emphasizes the unification of hierarchical decision-making within a single code-based framework, the chosen benchmarks and baselines primarily focus on high-level planning tasks. These environments do not inherently require dynamic adjustment of decision granularity, as most decisions occur at task-planning level. For instance, the ALFWorld case study in Appendix Figure 3 represents a classic high-level planning scenario where expressing actions as executable code offers limited additional benefit over traditional reasoning.

6. To substantiate the claimed benefits of unified decision granularity, ReCode would be evaluated in environments where both high-level planning and low-level control are essential such as robotic manipulation or embodied control tasks. In such settings, policies must integrate perception, planning, and motor control, providing a more realistic test of ReCode’s ability to adjust decision granularity. Moreover, conducting quantitative comparisons with recent frameworks like Code-as-Policies or Vision-Language-Action (VLA) models would more convincingly demonstrate the distinct contribution and practical value of ReCode’s recursive, code-centric decision mechanism.

**Questions:**

How is the mechanism for learning and controlling decision granularity concretely implemented and trained in ReCode?

---

> ### Author Response · Authors · 2025-11-21
> **Response to Reviewer SKxJ 1/2**
>
> We thank the reviewer for the constructive comments. Regarding the concerns of the reviewer SKxJ, we provide the following responses.
>
> > W1. & W2.
> > The paper does not sufficiently analyze or position ReCode against these methods in the related work section. Moreover, given the conceptual similarity, it would be valuable to include empirical comparisons with particularly REPL-Plan, one unifying reasoning and acting through executable code with adaptive feedback during execution. Technical novelties should be clearly specified.
>
> Thank you again for raising the connection to recent Code-as-Policies (CaP) and hierarchical code-generation approaches. In the revised version, we have added explicit discussion in **Section 2 Related Work** to clarify these distinctions (highlighted in blue).
> - CaP methods typically generate **a static program sketch** that is executed as a whole.
> - ReCode instead expands each placeholder **dynamically** and produces a full subtask-level plan in a single decision, conditioned **on the current observation**.
>
> This enables ReCode to re-decide the granularity at every recursive call, which static CaP frameworks cannot.
> We also clarified the difference from THREAD and REPL-Plan.
> - THREAD and REPL-Plan's recursive structure is mainly used for managing context and isolating subtasks **within a ReAct-style loop**.
> - ReCode produces **a complete plan** for each node at expansion time, forming **a unified top-down decision hierarchy** rather than a chain of local reactive loops.
>
> Together, these clarifications can more clearly position ReCode among existing code-based and recursive agent paradigms.
>
> > W5. & W6. ReCode would be evaluated in environments where both high-level planning and low-level control are essential such as robotic manipulation or embodied control tasks.
>
> We admit that our original experimental design did not fully consider environments requiring both high-level planning and low-level continuous control.
>
> Although the reviewers mentioned VLA-style baselines, our goal here is to show that ReCode also works for low-level continuous robot control, which can be tested without introducing visual perception; thus we keep the text interface and directly expose continuous control parameters.
>
> To address the concern, we added **ManiSkill manipulation experiments (PickCube, PullCube)** in **Section 4.4** using a text wrapper for continuous actions. As shown in the table below, ReCode clearly outperforms ReAct and CaP baselines, demonstrating that its unified recursive mechanism remains effective when both high-level planning and fine-grained continuous control are required.
>
> | Model            | Method | PickCube | PullCube |
> |------------------|--------|----------|----------|
> | **GPT-4o mini**  | ReAct  | 0        | 2        |
> |                  | CaP    | 34       | 34       |
> |                  | ReCode | 78       | 46       |
> | **Gemini 2.5 Flash** | ReAct  | 40       | 30       |
> |                  | CaP    | 82       | 44       |
> |                  | ReCode | 90       | 100      |

---

> > ### Author Response · Authors · 2025-11-21
> > **Response to Reviewer SKxJ 2/2**
> >
> > > W3. While the paper claims that flexible control of decision granularity leads to superior adaptability and efficiency, the current experiment results do not isolate granularity as a directly measurable factor. To substantiate this claim, ablation experiments varying the recursive depth could demonstrate how granularity impacts reward, cost, and performance–efficiency trade-offs.
> >
> > Thank you for the thoughtful suggestion.
> >
> > We clarify that ReCode is not a search-based method like MCTS, so it does not exhibit a cost–performance scaling trend driven by explicit search depth; however, it does involve a meaningful **performance–efficiency trade-off** through recursive depth.
> >
> > In the revised version, we added a controlled ablation varying the maximum recursion depth (**Fig. 3 in Section 4.2**). The results show a clear **inverted-U pattern**, indicating that decision granularity expressed through recursion depth has a direct impact on performance.
> >
> > > W4. Beyond qualitative case studies, the paper would present quantitative analyses that reveal how granularity dynamically changes during execution. Statistics such as the average depth of generated decision trees, branching factors, and the ratio of placeholder functions to primitive actions across different tasks would provide stronger empirical support for the proposed mechanism.
> >
> > We agree that the original submission lacked quantitative analysis of how granularity changes during execution. In the revised paper, we address this in two ways.
> > 1. We add a depth ablation in **Section 4.2**, showing that varying the maximum recursion depth directly affects performance, which provides evidence that granularity has measurable impact.
> > 2. We include in **Appendix A.4** a quantitative analysis of ReCode’s generated decision structures on ALFWorld, WebShop, and ScienceWorld. We report the average recursion depth, branching factor, and placeholder ratio, which together characterize how ReCode modulates coarse and fine decisions across tasks.
> >
> > > Q1. How is the mechanism for learning and controlling decision granularity concretely implemented and trained in ReCode?
> >
> > In ReCode, decision granularity is implemented and trained as follows:
> > - At inference time, the LLM is only called in the **Expansion** step: given a placeholder function, it generates a full code block that may contain new placeholders (coarse decisions) and primitive actions (fine decisions).
> > - This expansion is constrained by a prompt with few-shot examples that enforce the ReCode format, so the mix of placeholders and actions directly reflects the granularity chosen by the model.
> > - During training, we supervise this behavior by collecting Expansion input–output pairs from full recursive trajectories, so the model learns when to expand in a coarse way and when to commit to fine-grained actions.

---

### Author Response · Authors · 2025-11-21
**Public Comment 1**

We thank the reviewers for their helpful feedback.
We have substantially improved the paper, and all important revised content is marked in **blue**.
**Below is a concise summary of the key updates**:
1. (For Reviewer VZQm) We rewrote the contribution statements at the end of **Section 1 Introduction** to clearly articulate ReCode’s key novelty: unifying planning and action, introducing recursive code generation, and enabling multi-granularity training data.
2. (For Reviewer SKxJ, 23Bv) We expanded **Section 2 Related Work** by adding missing discussions on recursive agents, hierarchical planning, and code-as-policy paradigms. Particularly, we clarified how ReCode differs fundamentally from THREAD and REPL-Plan.
3. We redesigned **Figure 2** to explicitly separate the generation and execution flows, and added missing implementation details in **Section 3.3**, including task initialization, context management, error handling, and depth control.
4. (For Reviewer wfpU, VZQm) We added **AdaPlanner** and **ADaPT** in **Table 2** using Gemini 2.5 Flash and DeepSeek-V3.1 in addition to GPT-4o-mini, demonstrating that ReCode’s gains hold consistently across all baselines with different model families.
5. (For Reviewer SKxJ, 23Bv, wfpU) We added an ablation study on maximum recursion depth in **Section 4.2**, showing an inverted-U trend and justifying the chosen depth setting.
6. We added experiments analyzing data efficiency during supervised fine-tuning in **Section 4.3**, demonstrating that ReCode produces more compact yet more informative trajectories than ReAct.
7. (For Reviewer SKxJ, wfpU) We moved the original case-study examples to **Appendix A.5** and replaced **Section 4.4** with new experiments conducted in the **ManiSkill embodied manipulation environment**. These experiments verify that ReCode maintains advanced performance and robust decision-granularity control in continuous actions settings.
8. (For Reviewer SKxJ, 23Bv) We added runtime statistics in the appendix to better document ReCode’s runtime behavior and computational cost.

---

> ### Author Response · Authors · 2025-12-01
> **Public Comment 2**
>
> Dear Area Chair,
>
> We appreciate your time and effort in managing the review process under these unusual circumstances.
>
> We would like to briefly and concisely clarify two points regarding the status of our paper.
> First, we explain how the original reviewers adjusted their scores during the discussion phase based on our rebuttal before the leak incident occurred.
> Second, we summarize the additional experiments and revisions we have completed in direct response to the reviewers' core concerns.
>
> The following list summarizes the status of all reviewers before the rollback.
> - **Reviewer SKxJ** (Rating 4, Confidence 3) (No Reply)
> - **Reviewer 23Bv** (Rating 4, Confidence 4) (No Reply)
> - **Reviewer wfpU** (Rating 4 to 6, Confidence 4) (Updated Before Leak)
> - **Reviewer VZQm** (Rating 6, Confidence 3) (Questions Solved Before Leak)
>
> **Reviewer SKxJ (Rating 4, Confidence 3)**
>
> The reviewer did not respond during the discussion phase.
> The reviewer's main concerns focused on positioning against related work like Code-as-Policies and the lack of low-level continuous control experiments.
> The reviewer also requested quantitative analysis on decision granularity and ablation studies regarding recursive depth.
> We addressed the related work concerns by explicitly clarifying the distinctions in **Section 2** and further explained the differences in our response.
> We addressed the experimental scope by adding ManiSkill robotic manipulation tasks in **Section 4.4** to demonstrate unified low-level control.
> We addressed the analysis request by adding depth ablation studies in Figure 3 and detailed tree statistics in **Appendix A.4**.
>
> **Reviewer 23Bv (Rating 4, Confidence 4)**
>
> The reviewer did not respond during the discussion phase.
> The reviewer's main concerns focused on the novelty compared to prior recursive code generation works like REPL-Plan.
> The reviewer also requested quantitative measurements and analysis of decision granularity.
> We addressed the novelty concern by adding an explicit comparison with Code-as-Policies and REPL-Plan in **Section 2**.
> We addressed the analysis request by including depth ablation studies in **Section 4.2** to show the impact of recursion.
> We further provided structural statistics of the generated decision trees in **Appendix A.4** to characterize granularity.
>
> **Reviewer wfpU (Rating 4 to 6 on Nov 25, Strictly Prior to Leak)**
>
> The reviewer's main concerns focused on validation in continuous action spaces, error handling mechanisms, and additional baselines.
> We addressed the continuous control concern by adding ManiSkill manipulation experiments in **Section 4.4**.
> We clarified the error handling and failure mode analysis in **Section 3.3**.
> We also provided the requested AdaPlanner baselines on Gemini-Flash and DeepSeek in **Section 4.2**.
> The reviewer explicitly commented that their concerns had been addressed and raised the score based on these revisions.
>
> **Reviewer VZQm (Maintained Rating 6 on Nov 25)**
>
> The reviewer maintained the score of 6 and confirmed that all questions were solved.
> The reviewer requested clarification on the HeuristicConvert component and additional baselines.
> We renamed the component to avoid confusion and clarified the mechanism in **Section 3.3**.
> We also added the requested multi-model results for AdaPlanner in **Table 2**.
> The reviewer acknowledged our detailed response and confirmed that the questions were solved.
>
> Beyond specific reviewer requests, we implemented broader improvements to enhance the paper's quality and completeness.
> Please refer to our **Public Comment 1** for the full details of these revisions.

---

### Meta-Review · Area_Chair_sqsR · 2026-01-07

**Summary:**

Overall, the authors have addressed most reviewer concerns, such as lack of reference to existing work, granularity analysis, failure cases, and error recovery. The author has also added additional details about their framework to address clarification questions. Two reviewers explicitly acknowledged that their concerns were resolved.

However, since this paper positions itself around applications to robotics and planning, I believe the current experimental scope is still insufficient to fully justify the claimed use cases. While the added ManiSkill experiments help demonstrate feasibility in continuous control, they remain limited to relatively simple pick-and-place style tasks. To substantiate the broader claims about hierarchical planning and coding, the paper would benefit from additional evaluations on more complex robotic environments with richer hierarchical structure, or on long-horizon challenging benchmarks from the AI planning community.

Alternatively, if the authors intend to frame the contribution primarily as an agentic or code-centric paradigm, the current evaluations are also incomplete in that direction, as they do not include major LLM coding benchmarks or software engineering.

Overall, while the paper points toward an important and promising direction for integrating LLM code generation with interactive decision-making, the limited scope of evaluation leaves the central claims under-supported. Given these gaps, I recommend rejection.

**Reviewer Concerns:**

1. Additional references to prior work
2. Lack of granularity analysis and ablations (recursion-depth ablations and statistics)
3. Limited evaluation in text-based environments
4. Error analysis
5. Error recovery (from local failures, the new 3.3)

**Reviewer Scores:**

Reviewers wfpU and VZQm have explicitly responded that their concerns have been addressed.

Reviewer 23Bv asked for an abaltion study about the recursion depth, and the authors have included that in the rebuttal. Therefore this reviewer should be satisfied.

Reviewer SKxJ did not reply during the discussion period. Their main concern is about the applicability of the framework in actual robotics tasks, and they have suggested various baselines/references. I do not think the authors' added experiments are convincing enough from a robotic application perspective, given that the tasks are relatively simple (without hierarchical structures) and that the authors did not handle perception.

---

### Decision · Program_Chairs · 2026-01-26

Reject